

# An Estimate of Ice Wedge Volume for a High Arctic Polar Desert Environment, Fosheim Peninsula, Ellesmere Island

Claire Bernard-Grand'Maison[1] and Wayne Pollard[2]

[1]Department of Geography, Environment and Geomatics, University of Ottawa, Ottawa, K1N 6N5, Canada

[2]Department of Geography, McGill University, Montreal, H3A 0G4, Canada

*Correspondence to*: Claire Bernard-Grand'Maison (cbern085@uottawa.ca)

**Abstract.** Quantifying ground ice volume on a regional scale is necessary to assess the vulnerability of permafrost landscapes to thaw induced disturbance like terrain subsidence and to quantify potential carbon release. Ice wedges (IWs) are a ubiquitous ground ice landform in the Arctic. Their high spatial variability makes generalizing their potential role in landscape change

problematic. IWs form polygonal networks visible on satellite imagery from active layer surface troughs. This study focuses on the estimation of IW ice volume for the Fosheim Peninsula, Ellesmere Island, a continuous permafrost area characterized by polar desert conditions and extensive ground ice. We perform basic GIS analyses on high resolution satellite imagery to delineate IW troughs and estimate the associated IW ice volume using a 3D subsurface model. We demonstrate two semi-automated IW trough delineation methods with different strengths to increase time-efficiency of this process, done manually

in previous studies. Our methods yield acceptable IW ice volume estimates validating the value of GIS to estimate IW volume on much larger scales. We estimate that IWs are potentially present on 50% of the Fosheim Peninsula (± 3,000 km²) where 3.81 % of the top 5.9 m of permafrost could be IW ice.

## 1 Introduction

Arctic temperatures have increased twice as fast as the rest of the world over the past 50 years; a pattern that is expected to

continue for the next century (IPCC, 2013; AMAP, 2017). Permafrost is perennially cryotic ground that is estimated to underlie up to 20 % of the Earth's land surface (French, 2007) including vast areas in the Arctic that are threatened by climate change. Potential feedbacks of thawing permafrost include widespread landscape instability, accelerated coastal erosion and a massive release of carbon into the atmosphere, thus adding to the forcing on Earth's climate through the greenhouse gases effect (Schuur et al., 2015). The Canadian High Arctic permafrost is vulnerable to even a slight temperature increase because it lacks

insulation from vegetation and snow cover. Subsequent melting of ground ice reinforces the disturbance on permafrost's thermal equilibrium. These effects are already seen in the form of increased subsidence and rapid melting events (Pollard et al., 2015). Ice wedges (IWs), wedge shaped bodies of nearly pure ice, are a ground ice type ubiquitous in the High Arctic and areas of continuous permafrost in general. Investigating the response of IWs to climate change is a necessity to understand future permafrost degradation.



## 1.1 Ice Wedges and Thermokarst

The main factors controlling permafrost occurrence and depth of the active layer are: air temperature, vegetation type and cover, snow cover and topography (French, 2007). Thermokarst refers to all the processes related to permafrost degradation involving subsidence, erosion, collapse and instability resulting from thawing of ice-rich permafrost or the melting of massive

ice (van Everdingen, 1998). Ice-rich permafrost typically contains ice contents well in excess of the saturated moisture content of the host sediments; the volume of ice in excess of saturation is called excess ice. Thermokarst is initiated when the thermal equilibrium of permafrost is disrupted (i.e. increasing ground surface temperature); warming the upper part of the permafrost and increasing the depth of the active layer. Thermokarst alters surface hydrology favouring pond formation and gully formation, further causing permafrost erosion and thaw (e.g. Godin and Fortier, 2012). The magnitude of thermokarst driven

geomorphic change is highly dependent on ground ice content (Couture and Pollard, 2007; Kokelj and Jorgenson, 2013; Pollard et al., 2015). Thermokarsts can be geomorphologically significant in landscapes underlain by massive ground ice and IW ice. These permafrost features have the highest excess ice contents since they are usually composed of almost pure ice (Couture and Pollard, 1998) (Fig. 1).

Rapid cooling of ice rich soil can lead to thermal contraction and result in crack formation. At the beginning of the thaw season, contraction cracks are filled with melt water that freezes to form a vertical vein of ice. Over hundreds of years, IW growth occurs from re-forming cracks within existing IWs which adds a new ice veinlet in the permafrost (Lachenbruch, 1962). Intersection of IW cracks creates polygonal patterns that are widespread in the Arctic and for this reason has been the subject of abundant research (e.g. Lachenbruch, 1962; Black, 1976; Mackay, 1990).

Three types of IWs are identified based on their growth direction relative to the ground surface: epigenetic, syngenetic and anti-syngenetic wedges (Mackay, 1990). Epigenetic IWs typically grow on flat surfaces in pre-existing permafrost (Fig. 2). Their V-shape denote their tapered growth as cracks tend to form in the middle of the wedge. Syngenetic IWs grow upward as a response to surface aggradation of sediments. They are typically located in floodplains, where fluvial sedimentation occurs

and on the bottom of slopes, from the result of mass wasting. They are often nested in a chevron pattern. Anti-syngenetic IWs are characterized by a gradual downward growth pattern because of an incremental removal of surface material, for example on a slope affected by slow mass wasting. They penetrate deeper each year if thermal contraction cracking keeps pace with ice vein formation and their tops are truncated by thaw (Mackay, 1990). Epigenetic IWs are most typical and reflect a dynamic balance between climate and geomorphology where as the other types are less common and occur in areas of geomorphic

change (e.g. deposition and erosion). In this study, our analysis is concerned with epigenetic IWs. The distinct polygonal patterns produced by networks of IWs reflect the complex interaction between climate, materials and topography. In general, two polygonal morphotypes are recognized; namely high and low centred polygons depending on the microtopographic relationship between polygon troughs and polygon centres (Mackay, 2000).



As IWs grow, a shallow trough often develops over the ice body reflecting the quasi-stable relationship between the active layer and the IW top. In some cases, ridges parallel to the IW trough develop, creating low-centred polygons with raised rims often higher than 50 cm (French, 2007). Preferential thaw of IW polygons happens when snowmelt and runoff collect and flow in IW troughs creating a thermal perturbation. Changes in microtopography related to the evolution of high and low centred IW polygons plays an important role in surface process by influencing drainage, snow distribution and vegetation. These changes generate feedbacks that accentuate polygon morphology and eventually their vulnerability to thawing (Liljedahl et al. 2016).

## 1.2 High Arctic Climate Change

Climate change in the Arctic will impact the three most important environmental factors dictating permafrost and ground ice state, namely: air temperature, vegetation characteristics and snow cover. Regional interaction between these biological and physical dimensions make it difficult to generalize permafrost response to increased temperatures at the local scale. Decreasing albedo in the high latitudes due to decreased snow cover, glacier ice and sea ice extent affects atmospheric and oceanic circulation amplifying the increase in temperatures in the northern polar region (IPCC, 2013). In the Canadian High Arctic, increase in air temperature have largely resulted in winter and autumn warming and sites with little snow cover exposed to winds are therefore more responsive to changes in air temperatures (Smith et al., 2012). An increase in annual snow accumulation is also projected for cold high latitudes (AMAP, 2017). Since the 1980s, permafrost temperatures have risen in most regions, but this increase is particularly strong (0.4 °C to 1 °C per decade) in continuous cold permafrost regions such as the High Arctic (AMAP, 2017). This rise in ground temperature is disturbing the thermal equilibrium of permafrost beginning with changes in the active layer thickness and surface conditions. Increased thermokarst processes will enhance the permafrost carbon feedback through the release of greenhouse gases from a previously frozen organic pool (IPCC, 2013). The release of these large carbon and methane reserves will have an important effect on global temperatures on a millennial timescale (IPCC, 2013, Schuur et al., 2015). It is currently estimated that permafrost regions contain twice as much carbon that there is in the atmosphere (Schuur et al., 2015). However, the Canadian High Arctic permafrost is underrepresented in terms of datasets for carbon stocks (Brummell et al., 2014; Schuur et al., 2015). This region is dominated by a polar desert landscape (Walker et al., 2002) where the cold dry soils contains a smaller carbon pool than tundra and it is uncertain if it will stay as a carbon sink (Brummell et al., 2014; Schuur et al., 2015). No clear trend in increase in thermokarsts processes linked to climate change in the Arctic as been observed in the past decade (AMAP, 2017). However, widespread IW degradation has been observed in the Canadian Arctic Archipelago, Alaska and Siberia in recent decades and linked to climate change (Liljedahl et al., 2016). As climate change projections specific for the High Arctic polar deserts are poorly constrained due to a lack of data (Pollard et al., 2015), many uncertainties remain about their response to increase in temperature and permafrost stability.

### 1.3. Ice Wedge Volume Estimation

Analysis of permafrost and modelled disturbance due to an increase in ground temperature show that the "degree of response", specifically the magnitude of ground subsidence, is a direct function of the volume of excess ice (Couture and Pollard, 2007). Ground ice content is a key property to define permafrost terrain (Gogineni et al., 2014) and its estimation is necessary to

predict the sensitivity of a particular area to disturbance (Gilbert et al., 2016), which is important for engineering and environmental evaluations. It is also crucial for quantifying carbon pools and potential fluxes into the atmosphere (Kuhry et al., 2013; Ulrich et al., 2014). Understanding the cryostratigraphy and estimating ground ice type proportions also helps to reconstruct geomorphic history (Couture and Pollard, 2007; Gilbert et al., 2016). One of the first regional ground ice approximation study was performed by Pollard and French (1980) for Richards Island in the Mackenzie Delta, N.W.T.

Estimation of ground ice in the upper 10 m of permafrost was completed using drill log data, aerial photographs and topographic maps. Field intensive studies to characterize IW terrain and estimate ground ice content have recently been carried out in the Mackenzie Delta by Morse and Burn (2013) and on the Beaufort Sea coast in Alaska by Kanesvskiy et al. (2013). The lack of detailed field data for large and basically unstudied areas in the Arctic explains why ground ice distribution is often estimated at small regional scales.

Pore ice and segregated ice volumes can be determined from permafrost sample analysis, but not large ice bodies like wedges and massive ice. To specifically conduct IW volume estimations, knowledge of IW morphology as well as IW polygons geometry are required. Investigating IW morphology has been done effectively with ground penetrating radar (GPR) and electrical resistivity tomography (ERT) (e.g. Munroe et al., 2007; Bode et al., 2008; De Pascale et al., 2008; Léger et al., 2017).

Combination of such techniques with point measurement data (from exposed IWs, drilling, excavation and/or boreholes) helps to constrain IW type, shape and mean IW width and depth for a specific site (e.g. Pollard and French, 1980; Bode et al, 2008; Morse and Burn, 2013; Jorgenson et al., 2015). When lacking subsurface data, IW morphology is often approximated to be an inverted isosceles triangle and mean cross-sectional area of IWs is estimated with the use of width to depth ratios based on observations from exposed wedges. This works well for epigenetic and anti-syngenetic IWs but would underestimate

syngenetic ice volume due to their chevron pattern (Morse and Burn, 2013). Many other assumptions are often made which lead to over/under-estimation of wedge ice. For example, Kanevskiy et al. (2013) assumed that IW polygons were square and did not take into account the active layer thickness. Pollard and French (1980) adapted their IW volume estimation for areas with less developed polygons to come with an overall volume for Richard Island, N.W.T. Bearing in mind the assumptions made possibility yield large errors, such studies give some crucial information on relative proportion of IW volume as a first

approximation.

IW polygon geometry, mainly perimeter and total length of troughs for a defined surface area, can be manually calculated on a small scale from air photos (e.g. Pollard and French, 1980; Couture and Pollard, 1998). Improvements in remote sensing capabilities to do such task is needed to study the evolution of northern landscapes and to study ground ice (Jorgenson and





Grosse, 2016). In the review of recent advances in the study of ground ice by Gilbert et al. (2016), the use of satellite imagery has been recognized as the main contemporary method to determine IW polygon geometry on larger scales. Access to remotely sensed high-resolution imagery encouraged the development of techniques to measure polygon geometries using Geographic Information Systems (GIS). Ulrich et al. (2014) proposed a method to estimate IW volume from high resolution satellite images

and limited ground data using three-dimensional GIS tools. Polygon networks are delineated from satellite imagery and converted into a Triangular Irregular Network (TIN). Average IW width and depth from previous field surveys serve as inputs to a 3D subsurface model of polygons enabling wedge ice volume calculation. Their method was used in Yedoma deposits (Pleistocene-age ice-rich permafrost) and Holocene thermokarst basins in Siberia and Alaska. Two procedures were used to digitize the troughs centrelines in their study: manual delineation and Thiessen polygons delineation. Compared to previous

estimation methods, the volume accuracy is increased as the actual polygon dimensions are used. The study by Ulrich et al. (2014) establishes that GIS is an appropriate tool to conduct estimates of geometrically irregular features on a large scale but recognizes that precise field data is necessary.

IWs are the only massive ground ice phenomena associated with permafrost formation that occur throughout the Arctic that

can be mapped using high resolution satellite imagery. Previous studies have shown that IWs occur literally everywhere unconsolidated sediments are underlain by continuous permafrost and that their subsurface geometry is relatively consistent and closely related to terrain conditions and surficial geology (Couture and Pollard, 2007). Given the predicted changes in Arctic climate and our current understanding about nature and distribution of IWs, it is safe to assume that IW melt will contribute greatly to High Arctic geomorphic change with feedbacks that will reinforce permafrost instability. Manually

delineating IW polygons in satellite images as was done by Ulrich et al. (2014) is time consuming; semi-automated techniques to delineate polygons would greatly improve time efficiency and coverage are of wedge ice volume estimates. In this study, we estimate the volume of ice associated with IWs using a novel GIS approach in a specific region of the Canadian High Arctic, the Fosheim Peninsula. The goals of this study are twofold: (1) build upon the methodology introduced by Ulrich et al. (2014) by testing semi-automated methods to delineate IW trough and, (2) perform a rough estimation of IW ice volume of

the Fosheim Peninsula.

## 2 Study Area: Fosheim Peninsula

The Fosheim Peninsula lies within the Eureka Sound Lowlands (ESL), which includes the central part of Ellesmere and Axel Heiberg Islands in the north most part of the Canadian Arctic Archipelago. Our study focusses on the Fosheim Peninsula on Ellesmere Island which comprises roughly 70 % of the ESL region (Fig. 3). The Environment Canada weather station located

at Eureka (80°00′ N, 85°55′ W) is in the centre of the ESL on the Fosheim Peninsula. The area is flat to gently rolling, at elevations <200 m a.s.l. and the surface is comprised of mostly ice-rich silty-clay marine sediments underlain by continuous permafrost ~500 m deep. This area is characterized as a polar desert and is one of the driest regions in Canada, with a mean



annual precipitation of 68 mm recorded at Eureka for the period 1980-2015. The mean annual air temperature is -18.8 °C with the coldest mean monthly temperature in February of -37.4 °C and warmest in July of 6.2 °C for the same period. The thaw season varies between 3 and 6 weeks in length (Pollard et al., 2015). The mountains surrounding the ESL limit cold air masses from the ocean and create relatively warm July temperatures for this latitude and a general warming trend has been noted for

this month since 1980 (Pollard et al., 2015). The mean active layer thickness is 60 cm, and ranges between 30−100 cm. IW polygons are nearly continuous in unconsolidated sediments across the ESL and exposures of thick massive ice bodies are numerous (Pollard et al., 2015). It was estimated that wedge ice accounts for 1.8–3.5 % of total ground ice volume in this region, which is ±30.8 % of total ground volume in the upper 5.9 m of permafrost (Couture and Pollard, 1998). Average IW width is 1.46 m and depth 3.23 m from a survey of 150 exposed IWs by Couture and Pollard (1998). Extreme polar latitudes

often lack thermokarst features, but with ice content often exceeding 60–70 % in the fine marine sediments, the ESL is an exception (French, 2007; Pollard et al., 2015). A thin active layer with widespread ground ice make IWs in this region vulnerable to an increase in temperature. The response of High Arctic polar desert to projected climate change was modelled by Couture and Pollard (2007) with the climatic and geologic conditions of the ESL. They outlined two scenarios of +4.9 °C and +6.5 °C mean annual air temperature increase. These led to a lengthening of the thaw season by 26 days and increased

thaw depths of 17–20 cm. Comparison with modeled and past disturbance values reveal that ground subsidence is on the order of 1 m in the vicinity of IWs and greater than 1 m for massive ground ice bodies.

## 3 Methodology

### 3.1 Data Sources and Sample Areas

To assess the best techniques for IW trough delineation, we first identified a series of suitable sample areas from a detailed

analysis of four high resolution (0.5 m pixels) WorldView 2 and 3 satellite images. Like Ulrich et al. (2014) we defined the sample areas as squares of 250x250 m. Four sample areas, three on Ellesmere Island (EL1, 2 and 3) and one on Axel Heiberg Island (AH1), were selected with different polygon size, morphology, density and width of troughs (Table 1). All sample areas are characterized by random orthogonal polygons formed by epigenetic IWs on relatively flat surfaces (Fig. 4). The high-centred polygons on the Ellesmere Island sample areas have well-developed troughs (approximately 2–6 m wide). Wedge

hierarchy reflected by variability in trough width is most visible in sample area EL1. EL2 was chosen due to its dominance of rectilinear polygons, while EL3 was chosen for the high number of polygons with small areas and their proximity to polygons with much larger areas. In contrast, AH1 was chosen because of polygons with large areas and narrow troughs where cracking is assumed to be less frequent. AH1 is the only sample area where IW cracks not closing any polygons are visible.

### 3.2 Delineation of Polygons

At each sample area, the following three delineating methods were performed once using built in tools in ArcGIS (ESRI, Version 10.3.1): (1) Manual delineation, (2) Thiessen polygons and (3) Watershed Segmentation. In our method, we use and





refer to specific ArcGIS tools but most GIS packages contain similar tools and functions that could be used to replicate our analysis.

### 3.2.1 Manual Delineation

Following the Ulrich et al. (2014) methodology, we manually digitized polygons at each sample area by creating a line dataset
of the troughs centrelines. Only lines enclosing complete polygons falling within the sample area were kept and visible IW cracks not enclosing any polygons were also mapped.

### 3.2.2 Thiessen Polygons

The second method involved the semi-automated delineation of polygons based on the creation of Thiessen (or Voronoi) polygons. This approach was used at a few sites in Ulrich et al. (2014) to estimate volume of a relict IW network in
baydzherakhs landforms, where IWs had melted and only raised polygon centres remained. Thiessen polygons are defined mathematically as the perpendicular bisectors of the lines between all input points. Then, the area inside one Thiessen polygon is closer to its associated input point than any other input point (Aurenhammer, 1991). The tool "Create Thiessen Polygon" was used to create Thiessen polygons from manually chosen centre points of IW polygons, hereafter called the "approximated" centre points. Following this creation, polygons near the outer boundaries of the sample squares were necessarily defined by
having these boundaries as vertices. To avoid those edge effects, we created approximated centre points for polygons up to 30 m away from the sample areas before the creation of the Thiessen polygons. All resulting polygons that were not completely inside the sample areas were then deleted, and the remaining polygons were converted to a line dataset. Approximated centre points were created without the manual delineation lines visible to test the ability of the analyst to identify IW polygon centres.

### 3.2.3 Watershed Segmentation

Delineating IW polygons has many similarities with detecting grain boundaries in thin sections for petrographic analysis because both involve detecting edges. In a study by Barraud (2006) grain boundaries are detected using a "Watershed Segmentation" algorithm in image segmentation software. The basic principles of this segmentation process were reproduced in this study with the Spatial Analyst Hydrology toolbox of ArcGIS.

This third delineation method is based on the interpretation of the value of each pixel as a height function, i.e. as if it was a Digital Elevation Model (DEM). If the IW troughs have higher pixel values (brighter) than the polygon centres they will act as "mountains" and polygon centres as "valleys". If this topography was to be flooded, the water would accumulate in each polygon centre "valley" delineated by the trough boundary "mountains". In the WorldView images, the polygon centres have higher value pixels and the troughs lower value, therefore we inverted the pixel values before using the hydrology tools.



Watersheds were first obtained using the "Flow Direction" tool to calculate the flow direction of each pixel in the image, and then the "Basin" tool was used to delineate the smallest possible watersheds where water could accumulate. We converted the multiple steps of this method into a semi-automated process by implementing them in ArcGIS Model Builder (Fig. 5), which increased time-efficiency and required few interventions from the analyst. Filtering and smoothing of the image is required

before the "Flow Direction" and "Basin tool" outputs can provide watershed outlines that are representative of the IW polygons (Fig. 5a). To enhance troughs pattern, we used the "Focal Statistic Maximum" tool followed by the "Focal Statistic Mean" tool to reduce noise in the polygon centres. The later had to be performed multiple time to generate watersheds that were not over segmented, i.e. too many watersheds representing one actual IW polygon. Watersheds were created after each focal mean iteration and evaluated against the manually digitized polygons lines. The iterations were stopped when some watersheds

started to merge and to include two or more manually digitized polygons. At this point, approximately one to eight watersheds represented each actual IW polygon.

The hydrology tools were used on all of the available bands in WorldView imagery at each sample areas (Table 1) before being combined into a single line dataset per sample area. The same number of focal mean iterations were performed on all

15 the bands and they were then combined to create the final IW polygon delineation lines. For each band, watershed outlines were extracted as lines and were converted into a raster format (Fig. 5b). The three raster datasets of each band were summed, and pixels which were classified as boundaries (IW troughs) in two or more of the bands were kept (Fig. 5c). For these steps, the pixel size was increased from 0.5 to 1 m to get better chances of the watershed outlines overlapping. To convert the boundary pixel raster into a clean line dataset, the output raster was thinned, watershed boundary pixels converted to polylines,

and the "Extend Line" tool used with a maximum extension distance of 5 m (10 original pixels) to obtain a maximum number of closed polygons.

The clean trough centrelines datasets representing IW polygon outlines were visually assessed and edited to improve their accuracy. With the initial WorldView image visible, lines over-segmenting the IW polygons were deleted and lines were added

where polygons when some boundaries were not closed. All lines outside the sample areas were also deleted. Manually delineated polygons were included in these edited datasets to be consistent with the initial choice of the analyst. The remaining dangling lines were erased using the "Trim Line" tool. Finally, the "Simplify Line" tool with the point remove option and a tolerance of 1.5 m (3 pixels) was used to smooth any lines which had sharp edges due to the contouring of pixels form the conversion of raster to polyline format.

**3.3 Three-dimension Subsurface Model for Ice Wedge and Sediment Volume Calculation**

Similar to Ulrich et al. (2014) field data of mean IW depth and width was used to estimate IW volume, here from Couture and Pollard (1998). A buffer was created around the delineated lines of half the mean width of an IW (0.73 m) and then the buffer



extent was cut out of the polygons with the "Erase" tool. The resulting polygons therefore did not include the IW troughs. Every polygon, even the edge ones, is then considered to be surrounded by half an IW.

Volume calculations were performed in a 3D subsurface model by creating a Triangular Irregular Network (TIN) dataset, which is a network of mass points representing a surface terrain. Assuming that the IWs are inverted isosceles triangles, the elevation of 0 was assigned to the trough centrelines and an elevation of 3.23 m (mean IW depth) was assigned to the polygons. An elevation of 0 was also assigned to a dissolved polygon extent. The TIN was created from those three datasets with the Delaunay triangulation constrained for each segment (lines and polygon vertices) to be added as an edge in the TIN.

The IW volume and sediment volume were calculated using the "Surface Volume" tool. The IW volume was calculated from a plane above the TIN at 3.23 m. In order to compare the results of this study with the values found in Couture and Pollard (1998), the thickness of frozen soil considered in their study (5.9 m) was used to calculate sediment volume. It was calculated from a plane at -2.67 m below the TIN. The percent volume of IWs at each sample area was calculated by dividing the IW volume by the total frozen material (sediment and IWs) volume.

**3.4 Fosheim Peninsula Ice Wedge Volume Estimation**

We estimated the cumulative coverage area of IW polygons for the Fosheim Peninsula based on the surficial geology map from Bell (1992). The map was digitized with reference to the shoreline and contour datasets of CanVec series dataset from Natural Resource Canada (2016). As it is rare for IW polygons to occur in bedrock (French, 2007), it was assumed that they can be located only in unconsolidated surficial sediments (marine, fluvial, glacial sediments). The potential area occupied by
IWs was determined by subtracting the area of the large lakes and bedrock features. The 150 m CanVec contour was isolated as this provides a proxy for the Holocene marine limit on the Fosheim Peninsula because IWs are ubiquitous below this elevation (Bell, 1996; Couture and Pollard, 1998). We assumed that the mean of the IW percent volume of our sample areas was representative of the geomorphological settings on Fosheim Peninsula and used it to calculate the equivalent IW ice volume over the entire peninsula.

**4 Results**

**4.1 Delineation of Polygons**

To compare the accuracy of the two semi-automated delineation methods against the manual method, the mean perimeter and area of polygons (Fig. 6a) as well as the total length of delineated troughs (Table 2) were calculated. It is expected that a more efficient method would have a mean perimeter and mean centre point distance close to the manual delineation method values.



All the delineation methods provided polygon outlines for the four sample areas, although the accuracy of the outlines is variable. We assume that the Manual method provides the best outlines because troughs can be detected by the analyst regardless of their width and intersection with other troughs. Manually delineating the troughs was most difficult at EL3 where polygons were small and contrast was very low especially on the left side of the sample area (Fig. 4). Presence of IW troughs

that do not form closed polygons in AH1 were also difficult to detect as the troughs themselves were thin. When compared with the manual trough centrelines, the Thiessen polygons do not agree very well as they simplify the actual polygon shapes. Some edge effects remain on the Thiessen polygon boundaries, mostly caused by the proximity of polygons with large area difference (Fig. 4). The number of polygons were slightly increased at EL2 (+3.57%) and EL3 (+0.04%) and equal at EL1 and AH1 for the Thiessen polygons method (Table 2). The edited trough centrelines from the Watershed Segmentation method are

generally in good agreement with the manually digitized trough centrelines (Fig. 4). The Watershed segmentation technique overestimates the number of polygons, by as much as 5 times in the case of AH1 but around two times for the three Ellesmere Island sample areas (Table 2). This result is anticipated as watershed over segmentation is preferred to under segmentation before editing, because outlines of smaller polygons would disappear when watersheds would start to merge.

The mean distance between the centre points of the polygons for the different methods was calculated as an indicator of the similarity between the delineated polygons, particularly for the Thiessen approximated centre points versus the manual centre points (Fig. 6b). At all sample areas, this mean distance is <4 m, equivalent to <8 pixels. The maximum distance encountered was 9.5 m for a polygon on the edge of EL2 for the Thiessen polygons method. Two patterns in the mean distance between centre points seem to emerge: (1) the grouping of Manual/Thiessen Approximated with Manual/Watershed Segmentation and

of Thiessen/Thiessen Approximated with Manual/Thiessen due to their value closeness, and (2) the fact that the last group has higher values. The only exception to this observation is the Manual/Watershed Segmentation mean distance being the highest value for EL3.

The mean perimeter and polygon areas of the Thiessen and Watershed Segmentation methods have in majority a difference of

<5 % with the manual method at a given sample area (Fig. 6a). Exceptions occur at larger polygon sample areas with the Thiessen polygons method, where polygon area is overestimated by 11.6 % and 15.5 % for EL2 and AH1, respectively (Fig. 6a). Another exception occurs for mean perimeter of polygons at AH1, where it is underestimated by >10 % for each method (Fig. 6a). The Thiessen method overestimates the mean polygon area at each sample area, with proportionally greater overestimations for sample areas with larger polygons. The Watershed Segmentation area estimation is more precise, being

<1 % different than the mean area for the manually delineated polygons at all sample areas.

## 4.2 Ice Wedge Volume

An example of the TIN output for the IW volume calculation can be found in Fig. 7. All delineation methods included, the percent volume of IWs in the top 5.9 m of frozen material ranges from 1.41 %, for the lowest polygon density AH1, to 5.88 %





for the highest polygon density sample area EL3 (Table 2). IW volumes for the Watershed Segmentation and Thiessen methods are slightly lower or equal to the manual method estimate. The largest difference occurs at AH1 where there is a difference of -0.31 in the percent IW volume, which is equivalent at this sample area to 7.23 m$^3$ of IW ice. At each sample area, the IW volume estimate from the Thiessen polygons method is the lowest except in the case of EL1 where it is equal with the

Watershed Segmentation method, but still lower then the manual method estimates (Table 2).

Based on digitization of the Bell (1992) map, half of the Fosheim Peninsula surface area is potentially covered by IWs, corresponding to an area of ±3,000 km$^2$ (Fig. 8). Considering only the top 5.9 m of permafrost, this is equivalent to a volume of frozen material of 17.7 km$^3$. The total IW ice volume is $6.7 \times 10^8$ m$^3$, when assuming an IW volume of 3.81 % by averaging

the results from the manual delineation at the four sample areas (Table 2).

## 5 Discussion

### 5.1 Semi-automated Delineation Methods

The use of the Thiessen method on four sample areas with various polygon morphologies reveals its strength for volume estimation but not for trough identification. Indeed, the main problem with this method is that curved troughs could not be

delineated properly because the "Create Thiessen Polygon" tool can only output straight lines. This is reflected by the overestimation of polygon area (Fig. 6a). It is anticipated that better results would be obtained for hexagonal or rectangular polygonal patterns, rather than the orthogonal polygons tested in this study. This method is the least time consuming, but overall it underestimates IW volume by differing between 0.1-0.3 % from the percent IW volumes from manual delineation (Table 2). The Thiessen method was judged by Ulrich et al. (2014) to be "visually similar" to manual digitization. However,

their study areas had a majority of rectilinear polygons, for which the approximation of centre point is easier than for more complex shapes found in the sample areas used for this study.

The Watershed Segmentation method developed for this study with ArcGIS Hydrology tools was the most accurate in terms of locating trough centrelines and IW volume for every sample area. The poor agreement along the margins of sample area

EL3 can be attributed to the lack of contrast in this part of the image (Fig. 4), and explains why the mean distance between the automatically vs. manually derived centre points is the highest at this sample area (Fig. 6a). With minimal editing, the IW volume calculations using the Watershed Segmentation results were equal to the manual method values for two sample areas. The method accuracy can be improved by editing the sharp angles at boundaries that are not completely smoothed with the "Simplify Line" tool. These are most prevalent at sample area AH1 where the XY tolerance to simplify the lines (1.5 m) was

the smallest compared to the polygon vertices length. The accuracy of the trough centreline positions was reduced when increasing the pixel resolution to 1 m in the "Polyline to Raster" conversion but does not make a large difference in IW volume as the troughs are overwhelmingly larger than 1 m (2 pixels) at every sample area.



The effect of larger polygon size is visible in the results of sample area AH1 with the greatest differences in mean perimeter and area of polygons compared to manual delineation, and this independently of the method used. This can also be attributed to the thin troughs at this sample area and to the difficulty of differentiating what seems like dry runoff channels with IW

troughs (Fig. 4).

This study focussed on the development of two methodologies to delineate IW polygons based on only four sample areas. However, we are confident that both methods are applicable to delineating polygons for much larger areas. Thiessen polygons can readily be generated for larger areas and manually edited along the boundaries to reduce edge effects. The Watershed

Segmentation method can also be used for larger areas by choosing a number of focal mean iterations that will preserve the boundary details of the smallest polygons present. Even if over-segmented, this method preserves the largest polygon outlines corresponding to the darker zones of the images, interpreted as higher elevation when creating watersheds.

There are other semi-automated delineation methods that could be used to improve the delineation process of IW polygons on

satellite images. One potential technique is the method described by Li et al. (2008), which was used to delineate grain boundaries in rock thin sections. In this method, edge detection is based on the abrupt change in pixel values, representing brightness, at the boundary between two grains (or IW polygons). However, this method was deemed unsuitable since it requires considerable manual editing, and image classification algorithms could also not be applied due to the lack of contrast in some of our satellite images. Another approach would be to build on the methodology of Skurikhin et al. (2013) who

classified Arctic tundra drainage network components including IW troughs with image segmentation and shape-based classification. Various other image segmentation and classification techniques that are implemented in image processing softwares could be tested to delineate IW polygons on suitable imagery.

Another potential methodology would be using high resolution DEMs (>0.5 m horizontal and vertical accuracy) instead of

high resolution satellite images. Then, the watershed segmentation method developed in this study could be applied with more confidence. The need of higher resolution DEMs has also been identified for the study of permafrost degradation in general, to monitor surface subsidence and thermokarst processes (Jorgenson and Grosse, 2016). Promising remote sensing methods to detect topographic and subsurface change and to map ground ice distribution include: airborne light detection and ranging (LiDAR), interferometric Radar (InSAR), airborne ground penetrating radar and structure-from-motion technology (Gogineni

et al., 2014; Jorgenson and Grosse, 2016). High resolution terrain models can be derived from these methods that are needed to monitor surface subsidence at a smaller scale and to estimate ground ice cover over large areas (Gogineni et al., 2014). These data could be acquired from unmanned aerial vehicle (UAVs) or other airborne platforms and would require fieldwork. This highlights the strength of our relatively simple methodology that can be applied on remote locations without the need of





extensive fieldwork. Our goal was to build on the work of Ulrich et al. (2014) to develop a relatively simple methodology in a GIS that can be coded for semi-automation and that uses the resolution of the available data.

## 5.2 Ice Wedge Volume Calculations

IW volume derived from manual delineation at the four sample areas of this study are similar to the results of Couture and Pollard (1998) on the Fosheim Peninsula. Their study concluded that for "low density" polygonal terrain IW ice comprised 1.8 % of the top 5.9 m of permafrost and "high density" polygonal terrain 3.5 %. Their low density sample is very close to sample area AH1 (1.73 %), which confirms that the sample area on Axel Heiberg Island is representative of parts of the Fosheim Peninsula. However, our sample sites EL1 and EL3 have a much higher IW ice volume percentage, redefining "high density" polygonal terrain and associated IW volume percent on the Fosheim Peninsula. This may be due to the choice of sample areas of 250x250 m for estimating IW volume. This surface area was found by Ulrich et al. (2014) to provide a representative scale where polygon diameter showed only small variations. However, the polygon density and shapes of sites in Siberia used by Ulrich et al. (2014) may not be comparable to the sites tested here. In this study, this size was chosen as a manageable area for manual delineation and development of methodologies to delineate polygons, but the effect of scale of the extent considered on the IW perimeter, area and IW volume should be assessed to refine the IW volume estimation.

Multiple necessary assumptions are made when calculating IW volume with TINs and here we consider their potential effect in estimating IW volume at large scales. It was assumed that IW width and depth does not vary significantly between polygonal terrains, and lack of subsurface data means that using mean IW width and depth is the best approximation we can use for our calculations. However, one example where this assumption is not valid on the Fosheim Peninsula is in the surficial geology unit of thin veneer of glacial sediments identified by Bell (1992). The thickness of this geological unit over bedrock is defined as 2 m, which is less than the 3.23 m mean IW depth used here. This mean IW depth is a minimal estimate because only exposed IWs were measured by Couture and Pollard (1998). Like was done in their study, we used the depth of 5.9 m below the active layer to calculate the IW volume because no IWs were observed below this depth. The assumed fixed geometry of IWs as being isosceles triangles also impacts our IW volume calculation. It is the general shape of epigenetic IWs recognized by Mackay (1990) and the shape used in previous IW volume estimates (e.g. Pollard and French, 1980; Couture and Pollard, 1998; Bode et al., 2008; Ulrich et al., 2014). Even though IWs can be irregularly shaped in cross-section, an inverted isosceles triangle with its base corresponding to the IW width is the best approximation for shape for a calculation of this nature. For this study, it was assumed that all IWs in the Fosheim Peninsula were epigenetic. This may not be the case in areas characterized by high rates of sedimentation where syngenetic IWs may be present and lead to underestimation of IW ice volume. In the High Arctic, syngenetic IWs are limited to glacial forelands, alluvial fans and deltas.

IWs may contain gas inclusions, small amounts of sediment as disseminated grains and discontinuous veins of silt and fine sand (French, 2007). It was assumed in this study that they were composed of pure ice, as was also assumed by Ulrich et al.





(2014) and most previous studies (e.g. Pollard and French, 1980; Couture and Pollard, 1998; Bode et al., 2008). To verify this assumption would require a field sampling program to measure sediment content. Delineating IW polygons on satellite images implicitly assumes that all IWs have a visible surface expression (i.e. a trough structure). Field observations in the ESL show that this is not always the case (Pollard et al., 2015) (Fig. 3a) and would lead to an underestimation of IW volumes.

## 5.3 Impacts of Melting Ice Wedges

IWs are probably the most widespread ground ice phenomenon in areas of continuous permafrost. By virtue of their formative processes, the top of the IW usually corresponds with the base of the active layer. Since they are in a quasi-stable relationship with maximum seasonal thaw depth, then any increase in the active layer depth will result in a subsidence of the ground surface over the top of the IW. Over time, warmer summers produce small amounts of thaw at the top of the wedge leading to the formation of a shallow trough that marks the long axis of the wedge and emphasizes their polygon geometry. Under stable permafrost conditions, networks of shallow IW troughs will interact with snow distribution, surface vegetation and surface hydrology; in some cases, contributing locally to additional deepening and surface ponding. Hence, the localized degradation of IWs may be part of the normal evolution of permafrost landscapes. However, the widespread deepening of the active layer expected under projected Arctic climate change scenarios will lead to dramatic regional changes in landscapes marked by increased local topography and changes in surface hydrology (Liljedahl et al., 2016). As IWs melt out, the sides of the wedge will collapse into the open trough producing a highly dissected landscape characterized by mounds at the former polygon centres and networks of deep channels and shallow ponds along the former IW troughs (Couture and Pollard, 2007).

There is evidence that this is already beginning to occur on the Fosheim Peninsula (Fig. 3). An increase in thermokarst processes and retrogressive thaw slump retreat in the ESL over the past 25 years has been documented by Pollard et al. (2015). Unlike IW degradation associated mainly with thermal erosion by running water (e.g. Fortier et al., 2007) the instability of IWs in this region is related initially to thaw-induced surface collapse, undoubtedly running water will play a role at some point. The active melt out seen in Fig. 3b gives an indication of how rapid these changes may occur once the system becomes unstable. There is not only subsidence in the IW troughs, but also widespread back wasting of exposed IWs similar to the headwall retreat in a retrogressive thaw slump (Fig. 3c). There is also evidence of shallow active layer detachments along IW troughs in the ESL (Fig. 3b). In some cases, we have observed on the Fosheim Peninsula rapid melt out of IWs contributing to the formation of much larger retrogressive thaw slumps in areas where massive ground ice is present (Fig. 3a, d). The net result will be a period of landscape instability amplified by feedbacks associated with runoff (surface hydrology), snow accumulation, changing vegetation, thermokarst from massive ground ice, mass wasting, and microclimate. In principle, the new landscape will develop a deeper active layer consistent with the summer thaw conditions, though it may be long for the new active layer depth to stabilize, prolonging the period of thermokarst activity and subsidence. The new landscape will be quite different and depending on the topographic and geologic setting not unlike a badlands. For other areas, the new landscape



will reflect a geomorphic system affected not only by IW degradation but other changes to the permafrost system and surface hydrology.

## 6 Conclusion

IWs are one of the most common forms of ground ice in areas underlain by continuous permafrost. They exist in a quasi-stable equilibrium with seasonal thaw as defined by the depth of the active layer. Accordingly, a climate change driven increase in active layer depths will likely produce widespread instability of continuous permafrost landscapes associated with melting IWs. To better understand the potential impact of widespread destabilization of IW polygons there is a need to assess the volume and extent of IW ice. In the absence of detailed field observations, the analysis of IW polygons using high resolution satellite imagery and GIS based tools is the most logical solution. Based on our analysis of IW polygons for the Fosheim Peninsula we present three main conclusions. First, with minimal field data two semi-automated methods permit an acceptable approximation of IW volume in remote sample areas of the Arctic. The two GIS-based delineation techniques (Thiessen polygons and Watershed Segmentation) yield acceptable IW volume estimates compared to manual delineation as proposed in Ulrich et al. (2014). Implementation of these methods in a coded process accelerated the polygon delineation and demonstrates their potential to be applied to much larger areas in an efficient manner. Time constraint and required level of precision in the estimation of IW volume are two criteria to be considered when choosing one of these methods for future application in other sample areas. Second, IWs potentially cover an area of ±3,000 km$^2$ on the Fosheim Peninsula where 3.81% of the upper 5.9 m of permafrost is comprised of IW ice. We limit our calculation to the Fosheim Peninsula to correspond with available information on surficial geology; however we are confident that our results are applicable to the entire ESL. Further study in the ESL should focus on estimating IW volume for other sample areas using one of the semi-automated methods to increase the statistical significance of the results. Fieldwork in the ESL region could improve the IW volume estimates by linking surficial geology and physiographic units with IW polygon characteristics. Associated with other ground ice, carbon content estimation and field data, IW volume estimates will help to assess the vulnerability of High Arctic permafrost to climate change. Lastly, the occurrence of IWs increase the biophysical complexity of permafrost landscapes. Their widespread nature will contribute to significant permafrost instability once thermokarsts processes are initiated.

**Data availability**

The data used are listed in the references and tables and in the supplementary material.

**Competing interests**

The authors declare that they have no conflict of interest.



**Acknowledgments**

The authors acknowledge the support of ESRI Canada by awarding financial and technical support to C. Bernard-Grand'Maison to complete this project through the McGill University ESRI Canada Award 2016.This research was funded by the Canadian National Science and Engineering Research Council Discovery, Accelerator and Northern Supplement grants awarded to W. Pollard and a Polar Knowledge Canada Northern Scientific Training program grant awarded to C. Bernard-Grand'Maison. The writers would like to acknowledge the logistical support provided by the Polar Continental Shelf Program, Natural Resources Canada. The authors wish to acknowledge the field assistance, discussions and input of M. Templeton.

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



## Tables

**Table 1. Definition of the sample areas and satellite imagery used**

| Sample area | Image name, Date | Satellite | Sample area extent (top left corner) | No. of bands used (description) |
|---|---|---|---|---|
| **Ellesmere Island 1 (EL1)** | Slidre Fjord, 01/07/2012 | WorldView 2 | 80°00'28.7892'' N 85°50'28.5809'' W | 1 (Panchromatic Band) |
| **Ellesmere Island 2 (EL2)** | Slidre Fjord South 15, 08/09/2015 | WorldView 3 | 79°54'30.5496'' N 86°00'38.7151'' W | 3 (Pan-sharpened R, G, B) |
| **Ellesmere Island 3 (EL3)** | Slidre Fjord North 14, 08/09/2014 | WorldView 2 | 79°57'01.8072'' N 86°10'18.5485'' W | 3 (Pan-sharpened R, G, B) |
| **Axel Heiberg 1 (AH1)** | Axel Heibgerg 15, 29/07/2015 | WorldView 3 | 79°05'28.4936'' N 87°00'09.0523'' W | 3 (Pan-sharpened R, G, B) |



**Table 2. Summary of the delineation results for each method at each sample area and corresponding proportion of ice wedge volume**

| Sample area | Delineation method | Number of polygons (before edits) | Length of troughs (m) | Ice wedge volume (m³) | Frozen material volume (m³) | Proportion ice wedge volume (%) |
|---|---|---|---|---|---|---|
| | Manual | 164 | 6,220.44 | 12,753.25 | 280,401.53 | 4.55 |
| EL1 | Thiessen Polygons | 164 | 6,102.12 | 12,761.37 | 285,602.20 | 4.47 |
| | Watershed Segmentation | 164 (301) | 6,105.08 | 12,541.60 | 280,611.55 | 4.47 |
| | Manual | 56 | 3,457.68 | 6,905.93 | 224,451.24 | 3.08 |
| EL2 | Thiessen Polygons | 58 | 3,633.08 | 7,338.30 | 257,783.45 | 2.85 |
| | Watershed Segmentation | 56 (102) | 3,399.87 | 6,829.92 | 221,567.01 | 3.08 |
| | Manual | 271 | 7,896.53 | 16,515.05 | 280,986.34 | 5.88 |
| EL3 | Thiessen Polygons | 272 | 7,609.46 | 16,656.66 | 292,964.54 | 5.69 |
| | Watershed Segmentation | 271 (485) | 7,942.68 | 16,580.42 | 281,982.11 | 5.88 |
| | Manual | 9 | 1,568.67 | 2,603.85 | 150,828.45 | 1.73 |
| AH1 | Thiessen Polygons | 9 | 1,982.28 | 2,318.27 | 163,902.67 | 1.41 |
| | Watershed Segmentation | 9 (48) | 1,417.94 | 2,280.17 | 148,990.71 | 1.53 |



**Figures**

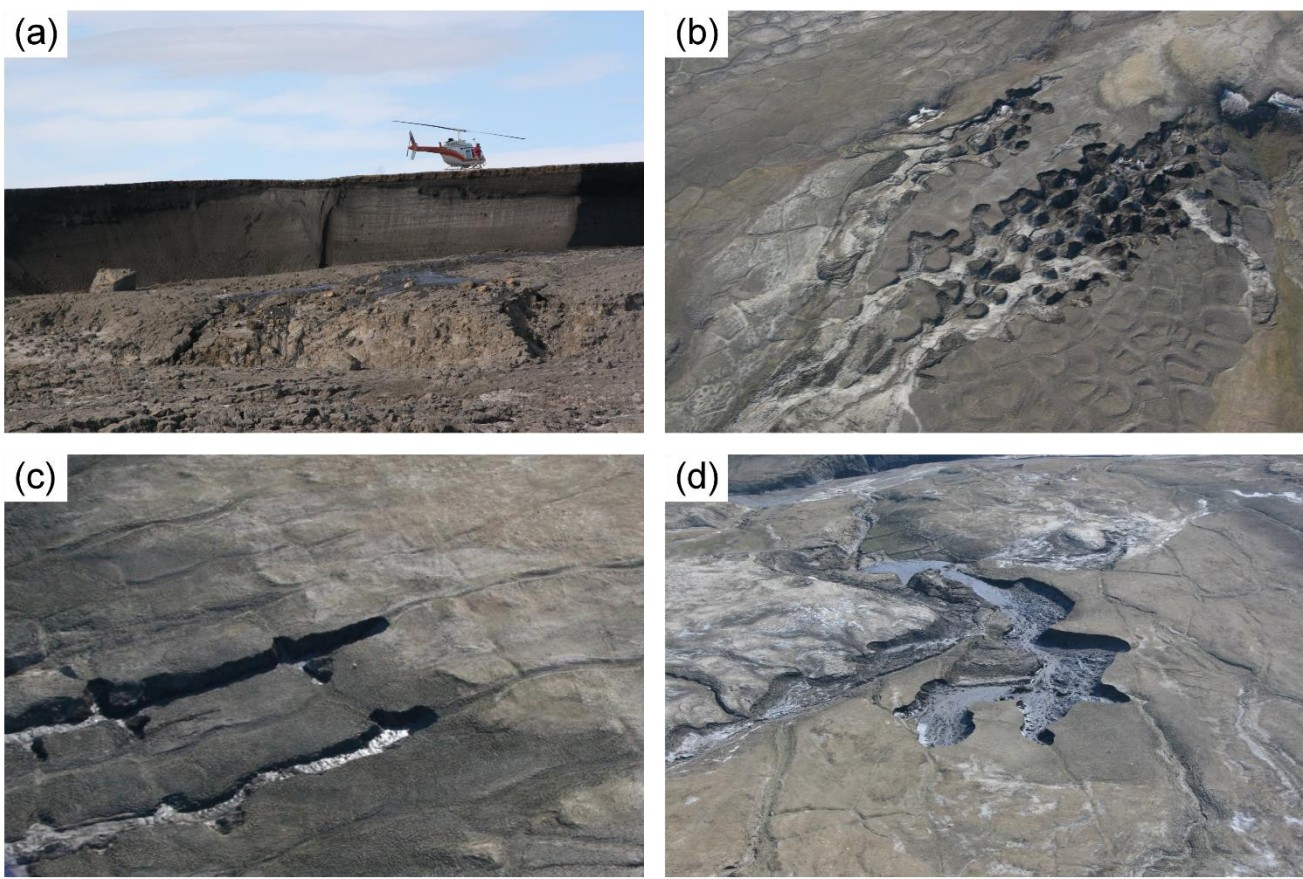

**Figure 1. Thermokarst processes in the Eureka Sound Lowlands. (a) Retrogressive thaw slump headwall with an exposed ice wedge**
5 **(~6 m length) with no surface expression, Axel Heiberg Island, July 2016. Helicopter and person for scale. (b) Aerial view of an active**
**melt out along ice wedge troughs and the resultant dissected landscape, Fosheim Peninsula, July 2015. (c) Example of back wasting**
**of ice wedges melting out, Fosheim Peninsula. July 2013. (d) Rapid melt out of ice wedges where massive ice is present, Fosheim**
**Peninsula July 2017.**



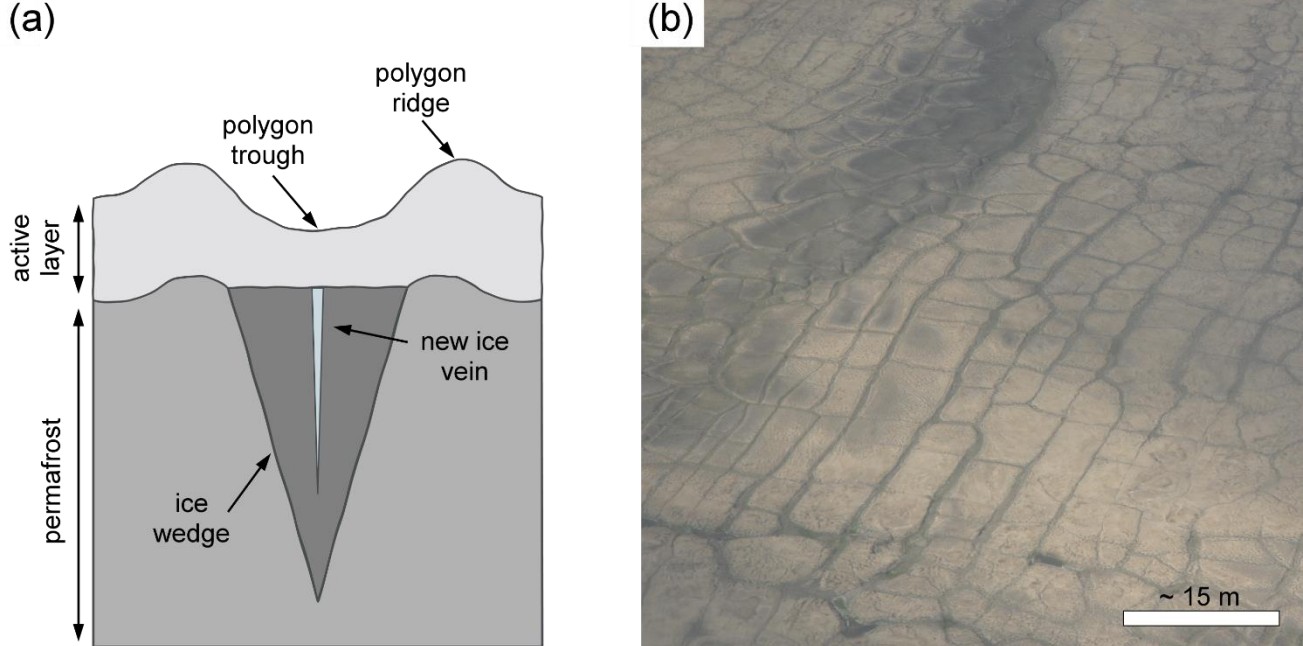

**Figure 2. Ice wedges surface expression. (a) Representation of an epigenetic ice wedge. (b) Aerial view of ice wedge polygons on the Fosheim Peninsula, Ellesmere Island.**





**Figure 3. Location of the Fosheim Peninsula in the Canadian High Arctic.**



**Figure 4. Original satellite image and delineation outputs with percentage of ice wedge volume in the top 5.9 m of permafrost for each method for each sample area.**



**Figure 5. Models developed with ArcGIS Model Builder for the Watershed Segmentation method. (a) Watershed creation: the inputs are the raster image as well as its maximum and minimum values needed to inverse the pixel values. Treatment of each site differed in the number of iteration the Focal Statistic Mean tool had to be performed for a satisfying basin segmentation output. The output is a basin raster, where every pixel has the value of its corresponding watershed. (b) Converting Basin output to a raster of the watershed borders: watershed boundaries are classified as a raster where the value of 1 represent boundaries. The snap raster is the initial band image clipped to the sample area. (c) Combination of the bands: all the classified watershed boundaries of each band are converted to a line feature that can be manually edited. Detailed description of the tools can be found at http://desktop.arcgis.com/en/arcmap/10.3/main/tools/a-quick-tour-of-geoprocessing-tool-references.htm**





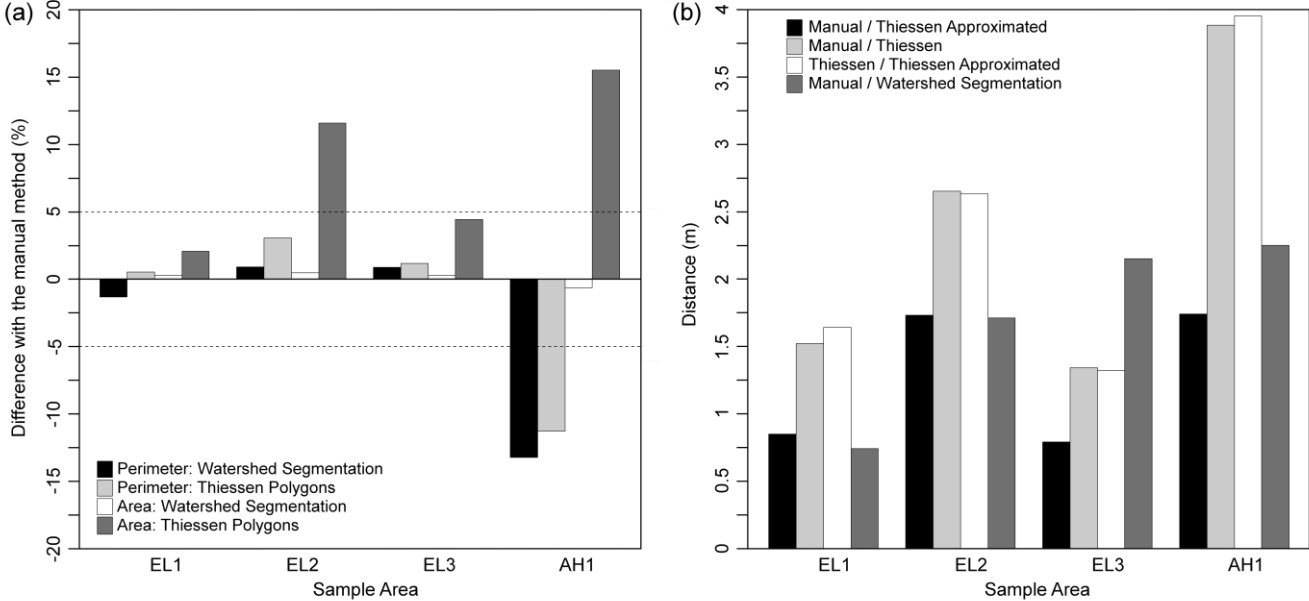

**Figure 6. Delineation method comparison metrics. (a)** Difference in mean area and perimeter of delineated polygons for the semi-automated methods with the manual delineation method. Refer to Table S1 for original data. **(b)** Mean distance between the centres of polygons created by the semi-automated delineation methods compared to the manual delineation method. The "Near" tool with a search radius of 10 m (20 pixels) was used. Two sets of centre points were considered for the Thiessen polygons method: the approximated centre point to create the polygons initially and the resulting centre points of the created Thiessen polygons. Refer to Table S2 for original data.




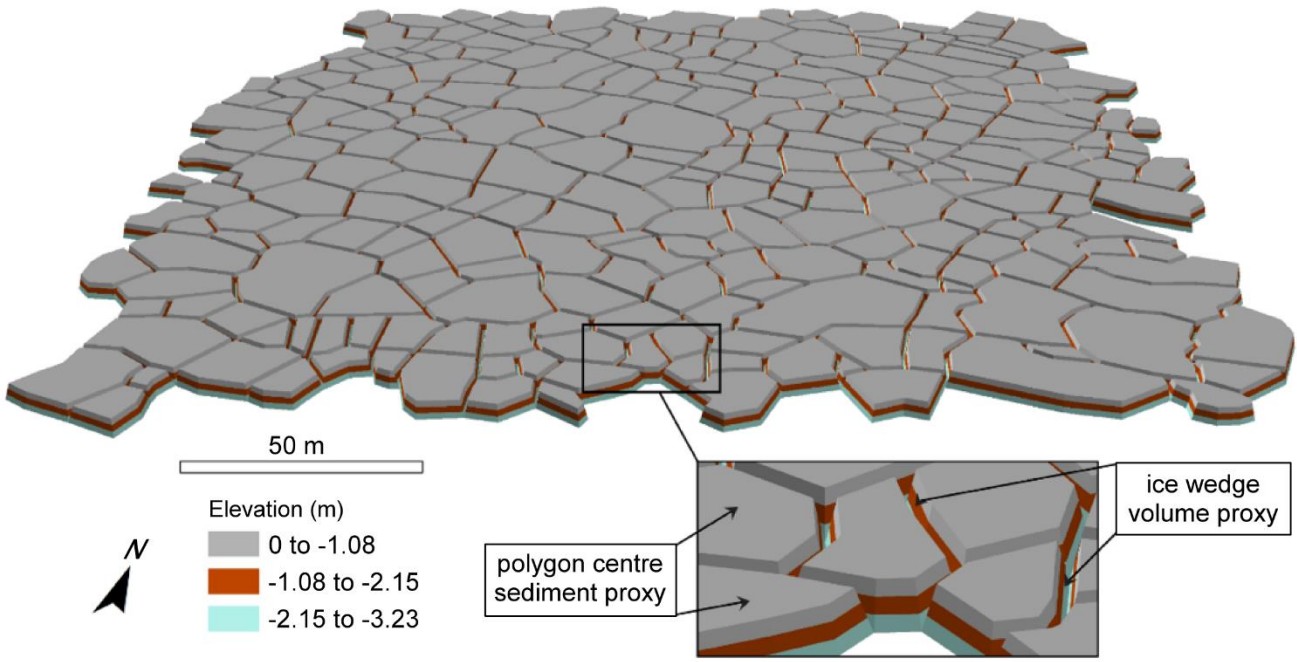

**Figure 7. Example of a 3D subsurface model. The TIN surface represents the entire EL3 sample area (250 m x 250 m). The origin elevation is at the bottom of the active layer.**



**Figure 8. Potential coverage area of ice wedges on the Fosheim Peninsula. Based on the surficial geology map produced by Bell (1992). The contours (CanVec data, Natural Resources Canada, 2016) are a proxy for the Holocene sea level on the Peninsula (Bell, 1996). Coordinate System: NAD 1983 UTM 16N. Projection: Transverse Mercator.**