# Peer review of "An Estimate of Ice Wedge Volume for a High Arctic Polar Desert Environment, Fosheim Peninsula, Ellesmere Island"

_The Cryosphere, 2018_

## Referee Comment (RC1) · Anonymous Referee #1 · 13 Mar 2018

Overall quality of the discussion paper ("general comments"): This paper by Bernhard-Graind'Maison & Pollard aims (1) to develop and test semi-automated GIS methods for mapping ice-wedge polygons by the delineation of ice-wedge polygon troughs and (2) to estimate the ice-wedge ice volume for the Fosheim Peninsula on Ellesmere Island in the Canadian High Arctic by using high resolution satellite imagery and 3D sub-surface models. Therefore they build upon a formerly published methodological GIS approach for ice-wedge volume calculations. The authors found that, in comparison to manual polygon trough delineation, their self-developed semi-automated polygon de-

lineation approach based on a watershed segmentation algorithm provides generally better results in polygon geometry and thus allows more accurate ice-wedge volume calculations than mapping by Thiessen polygons at their study sites. Finally, they provide amounts of ice-wedge coverage and volume for the Fosheim peninsula, which are pretty close to previous estimated amounts of ice-wedge ice volume. Even if this methodology-focused paper based in large parts on a formerly published methodological approach it provides new insights and data of ground-ice conditions in the Canadian High Arctic and new possibilities in semi-automated methods by simple GIS analyses for mapping polygonal networks on large scale. This is in particular an important issue in thinking about time-efficient automatic methods for a Circum-Arctic mapping of ice-wedge polygonal networks in relation to the better understanding of ground-ice conditions, permafrost landscape sensitivity to thaw and not at least the large-scale calculations of permafrost carbon stocks. Overall, I think that's a simple, short but nice methodological article. The simple and easily comprehensible GIS method is well presented in an easily accessible style. I have only a few suggestions and comments and can therefore imagine that this article will finally be published in The Cryosphere.

Individual scientific questions/issues ("specific comments"): P2/L1: Please include "soil type" or an adequate term in the list of main permafrost and active layer controlling factors. P2/L28-29: Do you talk about the High Arctic here? Please be more specific here. There are large areas in Siberia, for instance, in which syngenetic and epigenetic ice wedges equally widespread or even syngenetic ice wedges are representing the much larger proportion of ground ice. P6/L12-16: What time period is used to refer to the information given here? P6/L22: Here and/or even somewhere in the results, I miss information about the polygon sizes at the different study sites. I realized there are mean polygon areas in the supplement information but something related to polygon diameter and its variance at the individual sites would be good to have within the main text. P12/L6-12: I wonder what will be the effect on the watershed segmentation method of more complex contrast differences in satellite data, for instance, with regard to existence of water bodies in the center and the troughs typical for low-centered tundra ice-wedge polygons. I think more discussion about the applicability of the methods (especially the novel water segmentation method) beyond the Canadian High Arctic would be useful. General on section 5.2: If possible, I would like to see a little bit more discussion here about polygon size and geometry differences as well as ice-wedge volumes in relation to the site specific differences of the four study sites and vulnerability to thaw as done more generally in section 5.3. P13/L28-30: Please could you provide references here? P14/L3-4: And vice versa! There is not always a correspondingly large ice wedge below every crack and trough.

Technical corrections at the very end ("technical corrections": typing errors, etc.): P10/L2: Please change "Manual" to "manual" P10/L10: "Watershed Segmentation" is sometimes capitalized, sometimes small or one big and the other small. Please be consistent in spelling. See also the figures and captions. Figure 4 top: Please change "Thiesen" to "Thiessen" Figure 5a: Please change "Bassin" to "Basin"

---

## Referee Comment (RC2) · M. Kanevskiy (Referee) · 20 Mar 2018

This manuscript describes new methods of wedge-ice volume estimation based on GIS approach. It evaluates wedge-ice contents for the Fosheim Peninsula, Ellesmere Island, and compares various methods of such estimations. The paper will be very helpful for permafrost researchers who can use the suggested methods for estimations of wedge-ice volume in different permafrost regions, and I strongly support its publication. However, the manuscript needs some revision. My major comments and suggestions are listed below.

[Figure]

Page 1, Line 21: I recommend to cite a new edition of this book (French, 2018), where the numbers were updated (24%, see Table 5.1).

Page 2, Lines 21-30: Syngenetic IW are much more common than anti-syngenetic, and in some regions (e.g. yedoma regions in Siberia and North America) they occupy very large areas.

Page 4, Lines 17-22: It's better to start describing various methods of IW studies with exposures and drilling because geophysical methods are not very precise. Lines 21-22: Not all of the cited studies used this combination of techniques you're talking about.

Page 5, Lines 19-21: I recommend to cite previous studies because Ulrich et al (2014) already applied semi-automated technique in their study.

Page 5, Line 29: ESL region is not shown in Fig. 3.

Page 6, Lines 7-8: Please check these numbers. In the cited paper, it was stated that wedge ice accounts for 3.3% according to Table 2, and numbers 1.8 and 3.5 are from Table 1. Anyway, these numbers look confusing and I recommend to explain them better (probably you should compare them with volumes of other types of ground ice reported by Couture and Pollard 1998).

Page 9, Lines 10-14: I recommend to clarify your approach. It would be good if you add a simple equation or a figure.

Page 9, Lines 18-20: Actually, polygons in bedrocks are rather common in your study area. I recommend to explain your assumption in a different way: polygons may exist in bedrocks but wedge-ice volume is negligible.

Page 10, Lines 24-30: I recommend to report mean values (perimeter and/or area of IW polygons in different sample areas) either in the text or in Table 2.

Page 13, Line 8: I recommend to mention here that your values from EL2 are very similar to "high density" values in the cited paper.

Page 13, Line 16 – Page 14, Line 4: There are many publications on volumes of gas and solids in wedge ice (and their contents are rather small), so it is much more important for your purposes to obtain the field data on size and morphology of ice wedges specific for your sample areas – these numbers are very variable and may affect your IW volume estimations much stronger.

Page 14, Lines 11-13: I recommend to cite Jorgenson et al., 2006, 2015 here (they describe ice-wedge dynamics and related positive and negative ecological feedbacks).

Figure 4. Please change colors: it should be yellow for manual and blue for semi-automated methods.

Figure 7. This figure strongly resembles Figure 7 in Ulrich et al., 2014. I understand that you have already mentioned in the text that you follow their method (and this figure depicts your own sample area) but anyway I recommend to mention in the caption that this model was developed based on their approach.

MORE COMMENTS AND SUGGESTIONS ARE PROVIDED IN THE ATTACHED FILE.

Mikhail Kanevskiy Institute of Northern Engineering University of Alaska Fairbanks

Please also note the supplement to this comment:
https://www.the-cryosphere-discuss.net/tc-2018-29/tc-2018-29-RC2-supplement.pdf

**Supplement:**

[revised manuscript text omitted]

Three types of IWs are identified based on their growth direction relative to the ground surface: epigenetic, syngenetic and anti-syngenetic wedges (Mackay, 1990). Epigenetic IWs typically grow on flat surfaces in pre-existing permafrost (Fig. 2). Their V-shape denote their tapered growth as cracks tend to form in the middle of the wedge. Syngenetic IWs grow upward as a response to surface aggradation of sediments. They are typically located in floodplains, where fluvial sedimentation occurs, and on the bottom of slopes, from the result of mass wasting. They are often nested in a chevron pattern. Anti-syngenetic IWs are characterized by a gradual downward growth pattern because of an incremental removal of surface material, for example on a slope affected by slow mass wasting. They penetrate deeper each year if thermal contraction cracking keeps pace with ice vein formation and their tops are truncated by thaw (Mackay, 1990). Epigenetic IWs are most typical and reflect a dynamic balance between climate and geomorphology where as the other types are less common occur in areas of geomorphic change (e.g. deposition and erosion). In this study, our analysis is concerned with epigenetic IWs. The distinct polygonal patterns produced by networks of IWs reflect the complex interaction between climate, materials and topography. In general, two polygonal morphotypes are recognized; namely high and low centred polygons depending on the microtopographic relationship between polygon troughs and polygon centres (Mackay, 2000).

Probably you should also mention that in this study you analyze mainly high-centered polygons?

[revised manuscript text omitted]

I recommend to add more columns: mean perimeter and area of polygons, m²

[Figure]

**Figures**

[Figure]

**Figure 1.** Thermokarst processes in the Eureka Sound Lowlands. (a) Retrogressive thaw slump headwall with an exposed ice wedge
(~6 m length with no surface expression, Axel Heiberg Island, July 2016. Helicopter and person for scale. (b) Aerial view of an active
melt out along ice wedge troughs and the resultant dissected landscape, Fosheim Peninsula, July 2015. (c) Example of back wasting
of ice wedges melting out, Fosheim Peninsula. July 2013. (d) Rapid melt out of ice wedges where massive ice is present, Fosheim
Peninsula July 2017.

[Figure]

[Figure]

[Figure]

**Figure 2. Ice wedges surface expression. (a) Representation of an epigenetic ice wedge. (b) Aerial view of ice wedge polygons on the Fosheim Peninsula, Ellesmere Island.**

Most of ice-wedge polygons in your study area are high centered (see Figs. 2b and 4) but in Fig. 2a you show well-developed ridges typical of low-centered polygons.

[Figure]

**Figure 3. Location of the Fosheim Peninsula in the Canadian High Arctic.**

[Figure]

[Figure]

**Figure 4. Original satellite image and delineation outputs with percentage of ice wedge volume in the top 5.9 m of permafrost for each method for each sample area.**

I presume it should be yellow for manual and blue for semi-automated?

[Figure]

**Figure 5. Models developed with ArcGIS Model Builder for the Watershed Segmentation method. (a) Watershed creation: the inputs are the raster image as well as its maximum and minimum values needed to inverse the pixel values. Treatment of each site differed in the number of iteration the Focal Statistic Mean tool had to be performed for a satisfying basin segmentation output. The output is a basin raster, where every pixel has the value of its corresponding watershed. (b) Converting Basin output to a raster of the watershed borders: watershed boundaries are classified as a raster where the value of 1 represent boundaries. The snap raster is the initial band image clipped to the sample area. (c) Combination of the bands: all the classified watershed boundaries of each band are converted to a line feature that can be manually edited. Detailed description of the tools can be found at http://desktop.arcgis.com/en/arcmap/10.3/main/tools/a-quick-tour-of-geoprocessing-tool-references.htm**

[Figure]

[Figure]

[Figure]

**Figure 6. Delineation method comparison metrics. (a) Difference in mean area and perimeter of delineated polygons for the semi-automated methods with the manual delineation method. Refer to Table S1 for original data. (b) Mean distance between the centres of polygons created by the semi-automated delineation methods compared to the manual delineation method. The "Near" tool with a search radius of 10 m (20 pixels) was used. Two sets of centre points were considered for the Thiessen polygons method: the approximated centre point to create the polygons initially and the resulting centre points of the created Thiessen polygons. Refer to Table S2 for original data.**

[Figure]

[Figure]

**Figure 7.** Example of a 3D subsurface model. The TIN surface represents the entire EL3 sample area (250 m x 250 m). The origin elevation is at the bottom of the active layer.

This figure strongly resembles Figure 7 in Ulrich et al., 2014. I understand that you have already mentioned in the text that you follow their method (and this figure depicts your own sample area) but anyway I recommend to mention in the caption that this model was developed based on their approach.

[Figure]

**Figure 8. Potential coverage area of ice wedges on the Fosheim Peninsula. Based on the surficial geology map produced by Bell (1992). The contours (CanVec data, Natural Resources Canada, 2016) are a proxy for the Holocene sea level on the Peninsula (Bell, 1996). Coordinate System: NAD 1983 UTM 16N. Projection: Transverse Mercator.**

---

## Author Comment (AC1) · 29 Jun 2018

**Reply to interactive comments on**
**"An Estimate of Ice Wedge Volume for a High Arctic Polar Desert Environment, Fosheim Peninsula, Ellesmere Island"**
**by Claire Bernard-Grand'Maison and Wayne Pollard**

**Replies to Anonymous Referee #1**

**Overall quality of the discussion paper ("general comments"):**

This paper by Bernard-Grand'Maison & Pollard aims (1) to develop and test semi-automated GIS methods for mapping ice-wedge polygons by the delineation of ice-wedge polygon troughs and (2) to estimate the ice-wedge ice volume for the Fosheim Peninsula on Ellesmere Island in the Canadian High Arctic by using high resolution satellite imagery and 3D subsurface models. Therefore they build upon a formerly published methodological GIS approach for ice-wedge volume calculations. The authors found that, in comparison to manual polygon trough delineation, their self-developed semi-automated polygon delineation approach based on a watershed segmentation algorithm provides generally better results in polygon geometry and thus allows more accurate ice-wedge volume calculations than mapping by Thiessen polygons at their study sites. Finally, they provide amounts of ice-wedge coverage and volume for the Fosheim peninsula, which are pretty close to previous estimated amounts of ice-wedge ice volume. Even if this methodology-focused paper based in large parts on a formerly published methodological approach it provides new insights and data of ground-ice conditions in the Canadian High Arctic and new possibilities in semi-automated methods by simple GIS analyses for mapping polygonal networks on large scale. This is in particular an important issue in thinking about time-efficient automatic methods for a Circum-Arctic mapping of ice-wedge polygonal networks in relation to the better understanding of ground-ice conditions, permafrost landscape sensitivity to thaw and not at least the large-scale calculations of permafrost carbon stocks. Overall, I think that's a simple, short but nice methodological article. The simple and easily comprehensible GIS method is well presented in an easily accessible style. I have only a few suggestions and comments and can therefore imagine that this article will finally be published in The Cryosphere.

**Individual scientific questions/issues ("specific comments"):**

P2/L1: Please include "soil type" or an adequate term in the list of main permafrost and active layer controlling factors.

The sentence now reads: *"The main factors controlling permafrost occurrence and depth of the active layer are: air temperature, vegetation cover, soil type, snow cover and topography (French, 2007)."*

P2/L28-29: Do you talk about the High Arctic here? Please be more specific here. There are large areas in Siberia, for instance, in which syngenetic and epigenetic ice wedges equally widespread or even syngenetic ice wedges are representing the much larger proportion of ground ice.

Specific reference to the Canadian High Arctic polar desert has been added in this sentence.

P6/L12-16: What time period is used to refer to the information given here?

The sentence has been changed to explicitly write the time period considered from the work in Couture and Pollard (2007): *"They outlined two scenarios of +4.9 °C and +6.6 °C mean annual air temperature increase in the 2040-2060 period compared to mean annual air temperature from the 1948-1997 period."*

P6/L22: Here and/or even somewhere in the results, I miss information about the polygon sizes at the different study sites. I realized there are mean polygon areas in the supplement information but something related to polygon diameter and its variance at the individual sites would be good to have within the main text.

Following comments from Referee #2, the supplementary tables have been deleted and the mean perimeter and area of the polygons for each method and sites have been added in the main text to Table 2. The authors agree that it would be best to have a measure of the variance. However, all the GIS files have been lost due to a hard drive malfunction and the polygons would have to be re-digitized and would not be the same as when the volume was originally calculated. A general idea of variance can be visualized in Figure 4, where the satellite images of the sites are shown. The authors then judge that the information provided in the manuscript is sufficient.

P12/L6-12: I wonder what will be the effect on the watershed segmentation method of more complex contrast differences in satellite data, for instance, with regard to existence of water bodies in the center and the troughs typical for low-centered tundra ice-wedge polygons. I think more discussion about the applicability of the methods (especially the novel water segmentation method) beyond the Canadian High Arctic would be useful.

The authors acknowledge that the Watershed Segmentation methodology might not be applicable has it was described for terrain with prominent vegetation and low centered polygons with water bodies inside the polygon center and in the troughs. The methodology could be applied using high-resolution Digital elevation models (DEMs) on non High-Arctic terrain instead of the pixel brightness values. The paragraph has been extended to clarify this point and merged to the second to next paragraph about potential use of DEMs, see P12/L9-14.

General on section 5.2: If possible, I would like to see a little bit more discussion here about polygon size and geometry differences as well as ice-wedge volumes in relation to the site-specific differences of the four study sites and vulnerability to thaw as done more generally in section 5.3.

The authors feel that discussing geometry relative to ice wedge volume at this point in the paper would be premature. Site-specific geometry is not influencing the ice wedge volume calculation because the ice wedge sizes are assumed to be the same (mean width and depth found by Couture and Pollard (1998)). However, the density of polygons at each site influences the ice wedge volume as mentioned in the classification of "high density" and "low density" from Couture and Pollard (1998) but this is a qualitative description.

P13/L28-30: Please could you provide references here?
The authors argue that a reference is not needed because this sentence refers to field observations from W. Pollard. To reduce confusion the text has been changed to: *"Based on nearly 20 years of fieldwork on the Fosheim Peninsula, we have found syngenetic IWs relatively uncommon, and limited to areas of active sedimentation like glacial forelands (floodplains), alluvial fans and deltas. Assuming that all IWs in the Fosheim Peninsula were epigenetic should therefore not affect largely our IW volume calculation."*

P14/L3-4: And vice versa! There is not always a correspondingly large ice wedge below every crack and trough.
The authors agree and have now added this statement: "The opposite is also true as there might not be an IW below every crack and trough, but our estimates can only be based on what is visible in the satellite imagery."

**Technical corrections at the very end ("technical corrections": typing errors, etc.):**

P10/L2: Please change "Manual" to "manual"

Changed as suggested.

P10/L10: "Watershed Segmentation" is sometimes capitalized, sometimes small or one big and the other small. Please be consistent in spelling. See also the figures and captions.

A consistent spelling of *"Watershed Segmentation"* has been applied to the whole manuscript.

Figure 4 top: Please change "Thiesen" to "Thiessen"

Changed as suggested.

Figure 5a: Please change "Bassin" to "Basin"

Changed as suggested.

---

## Author Comment (AC2) · 29 Jun 2018

**Reply to Interactive comments on**
**"An Estimate of Ice Wedge Volume for a High Arctic Polar Desert Environment, Fosheim Peninsula, Ellesmere Island"**
**by Claire Bernard-Grand'Maison and Wayne Pollard**

**Replies to Referee #2: M. Kanevskiy**

**Overall quality of the discussion paper ("general comments"):**

This manuscript describes new methods of wedge-ice volume estimation based on GIS approach. It evaluates wedge-ice contents for the Fosheim Peninsula, Ellesmere Island, and compares various methods of such estimations. The paper will be very helpful for permafrost researchers who can use the suggested methods for estimations of wedge-ice volume in different permafrost regions, and I strongly support its publication. However, the manuscript needs some revision. My major comments and suggestions are listed below.

**Individual scientific questions/issues ("specific comments"):**

*Comments (questions and wording changes that we did not apply) from the supplement to the comments (https://www.the-cryosphere-discuss.net/tc-2018-29/tc-2018-29-RC2-supplement.pdf) have been added to this list and addressed.*

Page 1, Line 21: I recommend to cite a new edition of this book (French, 2018), where the numbers were updated (24%, see Table 5.1).

*The citation and the number have been updated according to French (2018). This edition was not available at the time of preparation of the manuscript.*

Page 2, Lines 21-30: Syngenetic IW are much more common than anti-syngenetic, and in some regions (e.g. yedoma regions in Siberia and North America) they occupy very large areas.

*We added reference to the Canadian Arctic for this sentence, where epigenetic IW are the most common.*

Page 2, Line 25: Accumulation of slope deposits

*Accumulation of slope deposit is the result of mass wasting. No changes were made.*

Page 2, bottom. Probably you should also mention that in this study you analyze mainly high-centered polygons?

*This information was added clearly as it was not mentioned in the description of the sites. Based on our observations high-centered polygons are the most common expression and typical of polar desert environments. To add clarity the sentence was changed to: "In this study, our analysis is concerned with epigenetic IWs, most commonly expressed as high-centered polygons in polar desert environments."*

Page 3. Section 1.2. Comment about decreased/increased snow cover.

*The decrease in snow cover is a generalisation for the high latitudes, see the globe. An increase in snow accumulation is projected for the High Arctic due to warmer winter temperatures and availability of moisture. The text was modified to improve clarity:*

*"Decreasing albedo due to diminishing snow cover, glacier ice and sea ice extent affects atmospheric and oceanic circulation amplifying the increase in temperatures in the northern polar region (IPCC, 2013). In the Canadian High Arctic, increase in air temperature has largely resulted in winter and autumn warming, and sites with little snow cover exposed to winds are therefore more responsive to changes in air temperatures (Smith et al., 2012). An increase in annual snow accumulation is also projected for high latitudes due to warmer temperatures (AMAP, 2017)."*

Page 4, Lines 17-22: It's better to start describing various methods of IW studies with exposures and drilling because geophysical methods are not very precise.

The order in which the methods are described has been changed in the paragraph.

Lines 21-22: Not all of the cited studies used this combination of techniques you're talking about.

The text was changed to make it clearer which studies are referred too for exposed IW and drilling and which use geophysical techniques:

*"Point measurement data from exposed IWs, excavation and/or boreholes helps to constrain IW type, shape and mean IW width and depth for a specific site (e.g. Pollard and French, 1980; Morse and Burn, 2013; Jorgenson et al., 2015). Geophysical techniques such as ground penetrating radar (GPR) and electrical resistivity tomography (ERT) have also been used to investigate IW morphology (e.g. Munroe et al., 2007; Bode et al., 2008; De Pascale et al., 2008; Léger et al., 2017)."*

Page 5, Line 14. What about pingos?

The text now reads: *"IWs and in some cases pingos are the only massive ground-ice types that can be mapped using high resolution satellite imagery."*

Page 5, Lines 19-21: I recommend to cite previous studies because Ulrich et al (2014) already applied semi-automated technique in their study.

The authors agree with this comment. The reference to Ulrich et al. (2014) was removed and the sentence was changed with no added references:

*"Semi-automated techniques to delineate IW polygons on satellite images would greatly improve time efficiency and coverage area of wedge ice volume estimates compared to manual delineation."*

Both techniques are already mentioned and described and referenced in the previous paragraph, so we see no need to repeat the references in this closing paragraph describing the objective of the study.

Page 5, Line 29: ESL region is not shown in Fig. 3.

The ESL region has never been clearly defined on a map. We modified the text to reference the Fosheim Peninsula which is shown in Fig. 3.

*"Our study focusses on the Fosheim Peninsula on Ellesmere Island (Fig. 3) which lies within the ESL region."*

Page 6, Lines 7-8: Please check these numbers. In the cited paper, it was stated that wedge ice accounts for 3.3% according to Table 2, and numbers 1.8 and 3.5 are from Table 1. Anyway, these numbers look confusing and I recommend to explain them better (probably you should compare them with volumes of other types of ground ice reported by Couture and Pollard 1998).

The authors agree that this sentence was confusing. After checking the numbers in the original Couture and Pollard (1998) paper the text was modified to: *"It was estimated that in this region wedge ice accounted for 1.8–3.5 % in volume of the upper 5.9 m of permafrost and that all types of ground ice combined accounted for 30.8 % (Couture and Pollard, 1998)."*

Page 7, Line 9. Use of Thiessen polygons in Ulrich et al. 2014: Not only!

The text has been modified to: "This approach was used in Ulrich et al. (2014) to estimate volume of a relict IW network in baydzherakhs landforms, where IWs had melted and only raised polygon centres remained, and at other sites to compare with manual delineation."

Page 9, Line 32. I'm not sure this is a good approach to attribute mean IW parameters from that study to all four sample areas that you use in this study because definitely these areas are very different: as you already mentioned, they have "different polygon size, morphology, density and width of troughs."

We acknowledge the reviewers concerns but are confident that our method adequately accounts for these differences. The three delineation techniques presented in the paper gives the centerlines of the IW through and not the width. Therefore, we don't see it justifiable to choose different width and depth qualitatively based on polygon size. We consider that applying our methodology to a large area is the best estimation we can provide at this stage.

Page 9, Lines 10-14: I recommend to clarify your approach. It would be good if you add a simple equation or a figure.

The authors acknowledge the reviewers comment but think that adding a figure or an equation is not necessary as the numbers provided are specific to our application and depend on the elevation reference used for the "Surface Volume" tool. All the given elevations in this paragraph and the previous have been changed to match the reference elevation of 0 m at the base of the active layer which corresponds to what was already indicated in Fig. 7 (now Fig. 6). Reference to this figure (Fig. 6, previously Fig. 7) was added to help the reader visualize the TIN and imagine the invisible planes used to calculate volumes. Clarifications in text have also been added concerning the provenance of the negative elevation values from mean IW depth and from depth of frozen soil considered.

Page 9, Lines 18-20: Actually, polygons in bedrocks are rather common in your study area. I recommend to explain your assumption in a different way: polygons may exist in bedrocks but wedge-ice volume is negligible.

We are of opinion that the reviewer is not entirely correct. Ice wedges in "solid" bedrock are not common in this area and as such bedrock areas can be ignored. By intact bedrock we mean areas of exposed rock occurring as outcrops in major ridges as defined in the surficial geology map by Bell (1992). The only ice wedges that might be considered as occurring in bedrock are where the bedrock is largely unconsolidated (Tertiary deposits) in which case their pattern is basically the same as areas that are discussed in this paper and are therefore included in our volume estimate. The sentence now reads: *"The potential area occupied by IWs was determined by subtracting the area of the large lakes and intact bedrock areas."*

Page 10, Lines 24-30: I recommend to report mean values (perimeter and/or area of IW polygons in different sample areas) either in the text or in Table 2.

Values of mean perimeter and area that are plotted as per cents in Figure 7 (previously Figure 6) have been added to Table 2.

Page 11, Line 10: I recommend to add these numbers to Table 2.

In this statement we were referring to the % IW volume of the manual delineation method that was found in Table 2. We think it would be confusing to add these in Table 2 as the averaging is only for the Manual method output. The text has been changed to: *"The total IW ice volume is $6.7x10^8$ $m^3$, when assuming an IW volume of 3.81 % by averaging the results from the manual delineation at the four sample areas in Table 2."*

Page 12, Line 21. Add references to "various other image segmentation and classification techniques"

This sentence was deleted from the text as the authors do not have sufficient knowledge of references in these fields and it was originally put as a suggestion for further studies.

Page 13, Line 1-2: The goals of your study are mentioned in the introduction and there is no need to mention them again but I recommend you to explain somewhere that unlike Ulrich et al. you have implemented one more method - watershed segmentation.

This is a valid point and the sentence was removed from the text here. It is now mentioned in the conclusion that the Watershed Segmentation is the new methodology presented in this study.

Page 13, Line 8: I recommend to mention here that your values from EL2 are very similar to "high density" values in the cited paper.

Changes has been made according to the suggestion: *"Even if our values from EL2 are very similar to the "high density" values in their study, EL1 and EL3 have a much higher IW ice volume percentage, redefining "high density" polygonal terrain on the Fosheim Peninsula."*

Page 13, Line 16 – Page 14, Line 4: There are many publications on volumes of gas and solids in wedge ice (and their contents are rather small), so it is much more important for your purposes to obtain the field data on size and morphology of ice wedges specific for your sample areas – these numbers are very variable and may affect your IW volume estimations much stronger.

The authors agree with this comment. Our ice wedge volume estimates are first approximations and that the variability in gas and sediment inclusion volumes as a factor in the estimate of ice wedge volumes is not realistic in a study of this nature. The text has been changed to emphasize this point:

*"IWs may contain gas inclusions, small amounts of sediment as disseminated grains and discontinuous veins of silt and fine sand (French, 2007). Inclusion of this factor in our volume calculation is not realistic for this first approximation study so it was assumed that IW were all composed of pure ice. This has also been assumed by Ulrich et al. (2014) and most previous studies (e.g. Pollard and French, 1980; Couture and Pollard, 1998; Bode et al., 2008)."*

Page 14, Lines 11-13: I recommend to cite Jorgenson et al., 2006, 2015 here (they describe ice-wedge dynamics and related positive and negative ecological feedbacks).

These references have been added to the specific sentence. However, since our research focuses on high latitude cold polar deserts the processes occurring in lower latitude warmer tussock tundra, (e.g. Alaska) are not entirely similar but nonetheless relevant.

Figure 1: What do you mean by length?

Length was changed to depth. It was meant ice wedge depth.

Figure 2: Most of ice-wedge polygons in your study area are high centered (see Figs. 2b and 4) but in Fig. 2a you show well-developed ridges typical of low-centered polygons.

The authors understand that from a certain perspective (there is no scale) this diagram could be confusing because of the surface of the active layer going down after the ridges, which is typical of low centered polygons. A small ridge can be seen in high-centered polygon in our area of interest but with no ponding in the middle of the trough, which would also be typical of low-centered polygons. The diagram and the caption have been modified as follows:

[Figure]

*"Figure 2. Ice wedges surface expression. (a) Representation of an epigenetic ice wedge in a high-centered polygon environment. (b) Aerial view of ice wedge polygons on the Fosheim Peninsula, Ellesmere Island."*

Figure 4. Please change colors: it should be yellow for manual and blue for semiautomated methods.

Changes made as suggested.

Figure 7. This figure strongly resembles Figure 7 in Ulrich et al., 2014. I understand that you have already mentioned in the text that you follow their method (and this figure depicts your own sample area) but anyway I recommend to mention in the caption that this model was developed based on their approach.

(Figure 7 is now Figure 6). Mention of Ulrich et al. (2014) was added in the figure caption.

**Technical corrections at the very end ("technical corrections": typing errors, etc.):**

In the supplement to the comments:

https://www.the-cryosphere-discuss.net/tc-2018-29/tc-2018-29-RC2-supplement.pdf

We appreciate the thorough comments of the reviewer on sentence structure and typing errors. Most of those changes have been applied to the manuscript and improve its clarity.

---

## Author Response (AR2)

**Reply to Editor comments on**
 ***"An Estimate of Ice Wedge Volume for a High Arctic Polar Desert Environment, Fosheim Peninsula, Ellesmere Island"***
**by Claire Bernard-Grand'Maison and Wayne Pollard**

**General comments:**

Dear Ms. Bernard-Grand'Maison,

Thank you for submitting your revised manuscript. Both reviewers recommended publication following some revision, and both gave you thoughtful comments that have improved the paper.

I agree with the reviewers that the manuscript should be published after some further revision. Most of the edits are minor and should be quick. I do have some comments that relate to the point raised by Reviewer 2 about your assumptions regarding the use of average ice wedge dimensions from Couture and Pollard (1998). Essentially, the paper argues that it is hard to justify different IW width and depth based on polygon size/trough length alone (agreed! There are so many reasons why there is no direct relation), so it is assumed that IW width does not vary significantly between polygonal terrain types. However, the problem is that Section 3.1, Figure 4, and Table 2 highlight 4 sample locations with different polygon sizes, morphologies, densities and, critically, widths. This makes it hard for a reader to agree with the assumption declared in Section 5.2. This section rightly points out that assumptions regarding IW width and depth affect volume estimates, but then the discussion focuses on implications regarding depth and does not treat width. The noted variability in width has implications on IW volume estimate and likely relates to sub-regional variation of geological setting/history (e.g., marine vs. fluvial vs. glacial).

In a general sense, it would be very informative to explore the implications stemming from assumptions regarding IW width variation, in addition to your exploration on trough length, as width and length are often both estimated for from remotely sensed images, whether the polygons are on Earth, Mars, or wherever. At minimum, it would be an important contribution simply to demonstrate the magnitude of variation that a range of assumed IW widths can have on regional volume estimates. It would give readers a sense of the margin of error, and can probably also be tied in to your discussion regarding the general assumption that all IW have a surface expression.

However, if your sample locations do suitably represent different geological contexts/settings, and you have some general surficial geology map data, it would be really great to tease out some aspects on sub-regional IW volume variation, and possibly refine the overall estimate of IW volume for the FP.

I think you can investigate this component of variation using the data already presented in the paper, and with out any additional digitizing. This is important as you indicated that the original polygon data are gone due to a hard drive failure.

This may take a bit of time to address, but when addressed should clear up and strengthen some of your arguments, and will demonstrate to readers the degree of sensitivity that IW volume estimates have to assumptions regarding IW dimensions.

My final comment is to see if you can improve the "hook". Your paper further develops the method for estimating IW length and I think this innovation can be better highlighted at the beginning to really grab the reader's attention. In the current version, your innovation isn't highlighted until close to the end of the paper.

I think that there will be a fair amount of interest in this paper, and I look forward to reading a revised version of the manuscript.

Detailed comments follow below.

Best regards,

Peter

Dear M. Morse,

Thank you for considering our manuscript for publication in The Cryosphere and providing detailed comments and edits to further improve the paper. We believe our revised version better highlights the new GIS methodology presented in the paper to "hook" the reader and makes the case for the value of our first order estimation of ice wedge volume for the Fosheim Peninsula. We have addressed the general comments by replying to the specific comments below (in red).

Regards,

Claire Bernard-Grand'Maison and Wayne Pollard

**Specific comments:**

P5/L14: Frost blisters and lithalsas are other massive ice types that can be mapped. You can go this route and cite a few more papers, or perhaps just say that IW often have a distinct surface expression that can be mapped using high resolution satellite imagery (cite appropriate references).

The authors agree that using the word "only" is misleading and ignoring the cases of frost blisters and lithalsas. The sentence was changed to: *"IWs often have a distinct surface expression that can be mapped using high resolution satellite imagery (Gilbert et al., 2016)."*

P5/L16: This statement is reason enough to explore (if possible) your volume estimates if you assume that your sites are representative of different surficial geological settings, rather than pooling them together.

The statement is: *"subsurface geometry [of ice wedges] is relatively consistent and closely related to terrain conditions and surficial geology (Couture and Pollard, 2007)."*

Looking at the original map by Bell (1992) from which the surficial geology data was taken to classify two regions (no ice wedges and potential ice wedge coverage) in Figure 8, all EL sites are from marine origin (gravel, sand, silt and clay deposited during higher sea level). The map only gives information for the Fosheim Peninsula, hence the surficial geology of site AH1 is not classified. This site is in a valley next to a large braided river and the origin of its surficial geology could be marine, glaciofluvial or fluvial. Our sample locations are then not representative of multiple surficial geological settings in terms of their origin (marine, fluvial, glacial or glaciofluvial sediments). Therefore, we cannot present a plausible estimate for each class where we could have differentiated between density of polygons, width and depth, which are probably related to their geological history. However, we think it is acceptable to pool all the classes together because the mean ice wedge width and depth that we use to estimate volume is representative of the glacial, fluvial and marine deposits as they are present within the surveyed area of Couture and Pollard (1998) centered on the Slidre Fjord. Any estimate of IW volume based on remote sensing techniques carried out over a large area with limited ground validation, which is what is presented in this paper for the Fosheim Peninsula, is as best a first approximation.

As the map by Bell (1992) is not accessible online for the readers to look up as complementary information, we decided to add more details concerning the surficial geology classes in the manuscript. In section 3.4, the classes are now properly listed in the first sentence and details is given on what classes have been included in the calculation of the area where ice wedges are present:

*"We estimated the cumulative coverage area of IW polygons for the Fosheim Peninsula based on the surficial geology map from Bell (1992), **which differentiates between surficial sediments of marine, fluvial, glaciofluvial and glacial origin as well as indicate weathered bedrock and residuum areas**. The map was digitized with reference to the shoreline and contour datasets of CanVec series dataset from Natural Resource Canada (2016). As it is rare for IW*

*polygons to occur in bedrock (French, 2007), it was assumed that they can be located **in all the unconsolidated surficial sediment classes (marine, fluvial, glaciofluvial and glacial).** The potential area occupied by IWs was determined by subtracting the area of the large lakes and areas identified as bedrock to the total area of the peninsula. The 150 m CanVec contour was isolated as this provides a proxy for the Holocene marine limit on the Fosheim Peninsula because IWs are ubiquitous below this elevation (Bell, 1996; Couture and Pollard, 1998). We assumed that the mean of the IW percent volume of our sample locations was representative of the geomorphological settings where IWs are present on Fosheim Peninsula and used it to calculate the equivalent IW ice volume over the entire peninsula."*

The definitions of the surficial geology classes have also been added to the caption of the new Figure 3, a combination of Figure 3 and 8. The legend of past Figure 8 will also be changed to reference the glaciofluvial sediments class which has been omitted due to its very small proportion. The new caption is:

*"Figure 3. Sample locations of this study and potential coverage area of ice wedges on the Fosheim Peninsula in the Canadian High Arctic. Surficial geology data is from a map produced by Bell (1992). The marine sediment class is defined as gravel, sand, silt and clay deposited during higher sea level. The fluvial sediments class is defined as gravel and sand deposited on floodplains and fans. The glacial sediment class is defined as non-sorted diamicton interpreted as till. The glaciofluvial sediment class is defined as gravel and sand deposited in the marginal zone of a former glacier. The 150 m contours (CanVec data, Natural Resources Canada, 2016) are a proxy for the Holocene sea level on the Peninsula (Bell, 1996). Coordinate System: NAD 1983 UTM 16N. Projection: Transverse Mercator."*

P5/L19-20: Cite Ulrich et al., 2014? They used Thiessen polygons for this reason.

The reference to Ulrich et al. (2014) was added.

P5/L23-24: Need a hook here too. Why do this? To develop a first order assessment of the response to climate change? Link to your conclusions.

The two last sentences of the paragraph were modified to include a "hook" to highlight the innovation of the paper (as mentioned in the general comments of the editor) and to justify why estimating ice wedge ice volume for the Fosheim Peninsula in a rough manner is relevant by making a link to section 5.3 Impacts of Melting Ice Wedges. The text now reads:

*"In this study, we build upon the methodology introduced by Ulrich et al. (2014) by testing GIS-based methods to delineate IW troughs and present a new semi-automated method based on watershed segmentation principles. We then provide a first approximation of IW ice volume in a High Arctic polar desert environment, the Fosheim Peninsula, to assess its sensitivity to thermokarst processes as a response to climate change."*

The presentation of a new delineation technique was also highlighted in the abstract, where a hook is most needed: *"We demonstrate the potential of two semi-automated IW trough delineation methods, one newly developed and one marginally used in previous studies, to increase time-efficiency of this process compared to manual delineation."*

P5/L27-28: If you can't map ESL is it because it is not well defined?
Suggest to change the text to: Our study focuses on the Fosheim Peninsula of Ellesmere Island (Fig. 3), which lies within the Eureka Sound Lowlands (ESL). The ESL include the central parts of Ellesmere and Axel Heiberg Islands in the northernmost part of the Canadian Arctic Archipelago.

The ESL has never been defined on a map but only qualitatively in a previous study by Pollard et al. (2015) by the geographic features bordering the regions (fjords and mountains). The text was changed to the suggested sentence and details on the area and bordering geographic features were added:

*"Our study focusses on the Fosheim Peninsula of Ellesmere Island (Fig. 3), which lies within the Eureka Sound Lowlands (ESL). The ESL region roughly covers 750 km$^2$ on central Ellesmere and Axel Heiberg Islands in the northernmost part of the Canadian Arctic Archipelago and is bordered by the Sawtooth Mountains to the east and the Mueller Ice cap to the west (Pollard et al., 2015).*

P5/L29: Before this text, please include a line or two about the general geology and unconsolidated surficial materials that you later use to make your regional IW volume estimation (Fig. 8 indicates marine, fluvial, and glacial sediments are present).

Details on the surficial geology from Bell (1996) have been added:

*"Surficial geology of the area is mainly composed of unconsolidated sediments of glacial, fluvial and marine origins. The area is mostly flat to gently rolling with elevations below 300 m a.s.l. except for several ridges which rises to a maximum of 840 m a.s.l. where outcrops of intact bedrock occur (Bell, 1996). Ice-rich silty-clay marine sediments dominate below the ~150 m a.s.l. Holocene marine limit (Bell, 1996) and are underlain by continuous permafrost ~500 m deep (Pollard et al., 2015)."*

P6/L18: Search and replace "Sample Areas" with "Sample locations". Also check for this in figures and tables. The areas are all the same, the locations are different.

Changes made as suggested.

P7/L1: Reviewer 1 asked for some indication of the variance of polygon dimensions. Recognizing that your GIS files are gone, can you please make a note that somehow indicates that while the importance of assessing variance is recognized, it couldn't be treated quantitatively, but can assessed quantitatively by examining Figure 4.

The authors agree that recognizing that variance would have been an interesting parameter to compute is a good idea. A sentence was added at the end of the first paragraph in the results section 4.1 after the mention of Fig. 7a and Table 2 where variance would have been added. Added sentence:

*"While we recognize the importance of assessing variance in polygon size at the sample locations, it couldn't be treated quantitatively but can be assessed qualitatively by examining Figure 4."*

P11/L16: 2 other methods were employed. What are the other volumes according to the semi-automated techniques? If the differences are small, it would add strength to the argument that semi-automated techniques are an improvement. This may also allow you to point out in your discussion that more error may be introduced by width assumptions than by the technique used to determine IW length.

The volumes for the two other methods have been added to this paragraph. The differences are indeed small and both volumes are lower than for the manual method. The text now reads: *"The total IW ice volume is 6.7x10$^8$ m$^3$, when assuming an IW volume of 3.81 % by averaging the results from the manual delineation method at the four sample locations in Table 2. Slightly lower estimates are obtained when averaging the IW volume of the two other semi-automated methods: 6.4x10$^8$ m$^3$ with 3.61% for the Theissen method and 6.6x10$^8$ m$^3$ with 3.74% for the Watershed Segmentation method."*

P12/L31-32: Please clarify this statement. Some of the ideas in this paragraph are a bit jumbled together. On line 22 you say that the method should be applied to a DEM to gain more confidence, but later in the same paragraph you indicate that high-res. DEM generation requires field work. Then here you say that you don't need field work.

The authors agree that this sentence might be confusing. The methodology we present is based on satellite images only and not on DEMs. Creating these DEMs with the techniques mentioned in the previous lines would require fieldwork. The only fieldwork required in our methodology is to get an average width and depth if this cannot be

found in the published literature. To clarify, the sentence was changed to: *"This highlights the strength of our relatively simple methodology, relying on high resolution satellite images and minimal field data, which can be applied on remote locations without the need of extensive fieldwork to create DEMs."*

P13/L26: Implications for gross error regarding assumed IW width should also be discussed. Differences in width are obvious in the images.

The average width estimate from Couture and Pollard (1998) is based on a small sample of measurements and the authors argue that it would be misleading to offer any amount of error analysis. When attempting estimates on the regional scale considered in this study, it is necessary to make assumptions that reduce the absolute accuracy of the calculation.  We are confident that our total ice wedge volume calculation for the entire peninsula is a reasonable "low" estimate.  However, given the limitation of 3 variables already discussed in Section 5.2 (assumptions of mean width, mean depth and representative surficial expression), we modified the manuscript to refer to our estimate as a "first approximation".  It is also not possible to provide a range of IW estimates for the entire Peninsula using different width and depth without additional digitizing and without the original files which have been lost in a hard drive failure. We used a TIN to calculate the total frozen sediment volume and the ice wedge volume at each site and this is constrained in three dimensions by the delineated ice wedge trough lines which we do not have anymore. Therefore, we cannot simply apply a new width and depth to the total ice wedge length in each site for the manual method because we do not have the volume of total frozen sediments to calculate a new percentage of IW. For these reasons, we argue that a new calculation would not be possible, nor would it increase the accuracy of the estimate. We think that the assumptions made are valid and have modified section 5.2 to emphasise these points. We are confident that there is value in presenting this first approximation to inform on the magnitude of landscape change that can be expected at a regional scale in a polar desert environment where ice wedges are omnipresent. Here is the main text modified in Section 5.2 concerning ice wedge width:

*"Multiple necessary assumptions are made when calculating IW volume with TINs and here we consider their potential effect in estimating IW volume at large scales. The most critical is probably the assumption that IW width and depth does not vary significantly between polygonal terrains, and lack of subsurface data meant that using mean IW width and depth was the best approximation we could use for our calculations. The small variability in estimations of IW volume for the entire peninsula from the three delineation methods suggests that more error might be introduced in our estimate from the assumption of a fixed IW geometry than by the technique used to derive IW length in a specific area. Differences in IW width at our sample locations are obvious in Figure 4 where multiple troughs are greater than the 1.46 m average used (e.g. EL2) and likely relates to sub-regional variation in geological history. To use these visible differences in surface expression as information of IW width would require another assumption that cannot be validated with the limited field data available: that trough width approximate IW width. Multiple sample locations in each surficial geology class presented by Bell (1992) would have permitted the calculation of IW volume on a sub-regional basis and the definition of IW parameters such as apparent width and polygon density for each surficial geology class. A specific example where assuming a fixed IW geometry is not valid on the Fosheim Peninsula is in the surficial geology unit of thin veneer of glacial sediments identified by Bell (1992). The thickness of this geological unit over bedrock is defined as 2 m, which is less than the 3.23 m mean IW depth used here. It is important to mention that the IW width and depth used in our calculations are minimal estimates because only exposed IWs were measured by Couture and Pollard (1998). Like was done in their study, we used the depth of 5.9 m below the active layer to calculate the IW volume because no IWs were observed below this depth."*

The justification of our estimate "as a first order approximation" has been added to conclude section 5.2: *"Given the potential errors discussed above associated with assumptions of width, depth and surface expression of IWs on the Fosheim Peninsula, we refer to our estimate of total IW volume as a "first approximation" and are confident it is a reasonable minimum estimate for this regional scale."*

Our abstract and second point in the conclusion was also modified to position our estimate as first approximation:

*"Secondly, IWs potentially cover an area of ~3,000 km² on the Fosheim Peninsula where a minimum of 3.81% of the upper 5.9 m of permafrost is comprised of IW ice. This first approximation is based on limited field validation data and sample locations which constrains it to the Fosheim Peninsula; however we are confident that our results are applicable to the entire ESL."*

P14/L12: Indicate why. Inactive? Incipient? obscured by mass wasting (Anti-syngenetic? Syngenetic?)?

Information from the cited source was added to the sentence to indicate why some IW do not have a visible surface expression in the ESL: *"Field observations in the ESL show that this is not always the case because many of the factors leading to trough development (e.g. vegetation coverage and surface hydrology) do not always apply in very cold and relatively dry polar desert environments. Usually, no trough structure is visible when the top of an IW is in equilibrium with the thin active layer depth (Fig. 1a) (Pollard et al., 2015). This assumption would lead to an underestimation of IW volumes on the Fosheim Peninsula."*

P14/L14: You use 3 methods to delineate the polygons, only one is visual inspection. Suggest to delete the word "visible".

Changed as suggested.

P16/15: Please use a consistent style. For example, some titles are abbreviated and others are not.

A consistent style was applied to the references.

Figure 1: Possibly an anti-syngenetic ice wedge [in a)]? Interesting if an epigenetic wedge of that great size did not have a trough/surface expression.

The ice wedge shown in a) was found in a stable plain terrain and is believed to be epigenetic in nature. This detail was added in the caption: *"Figure 1. Thermokarst processes in the Eureka Sound Lowlands. (a) Retrogressive thaw slump headwall with an exposed epigenetic ice wedge (~6 m depth) with no surface expression, Axel Heiberg Island, July 2016. Helicopter and person for scale. (b) Aerial view of an active melt out along ice wedge troughs and the resultant dissected landscape, Fosheim Peninsula, July 2015. (c) Example of back wasting of ice wedges melting out, Fosheim Peninsula. July 2013. (d) Rapid melt out of ice wedges where massive ice is present, Fosheim Peninsula July 2017."*

Figure 2: By definition a high centred polygon does not ridges because they have collapsed (the centre is now high).

The authors agree with this comment. The center of the polygon on the figure was raised due to comments of Reviewer 2 that the majority of the ice-wedge polygons in our study area are high centered and diagram represented a typical low-centered polygon with ridges. The description "polygon ridge" has been changed to "polygon centre", which is an important term in the manuscript. Updated figure and caption below:

[Figure]

*"Figure 2. Ice wedges surface expression. (a) Representation of an epigenetic ice wedge in a high-centered polygon environment. (b) Aerial view of ice wedge polygons on the Fosheim Peninsula, Ellesmere Island."*

Figure 3: Please show sample locations in Figure 3. From a geographic perspective, the most useful figure is Figure 8, while these two maps provide context. Move Figure 8 here and use these two as insets. This will lower the figure count and you can then reference Fig. 3 as a part of your new background info on geology.

Figure 8 was merged with Figure 3 as suggested. The new Figure 3 is referenced as part of the geologic background info in section 2 Study Area: Fosheim Peninsula. New figure and caption below:

[Figure]

*"Figure 3. Sample locations of this study and potential coverage area of ice wedges on the Fosheim Peninsula in the Canadian High Arctic, shown in inset. Surficial geology data is from a map produced by Bell (1992). The marine sediment class is defined as gravel, sand, silt and clay deposited during higher sea level. The fluvial sediments class is defined as gravel and sand deposited on floodplains and fans. The glacial sediment class is defined as non-sorted diamicton interpreted as till. The glaciofluvial sediment class is defined as gravel and sand deposited in the marginal zone of a former glacier. The 150 m contours (CanVec data, Natural Resources Canada, 2016) are a proxy for the Holocene sea level on the Peninsula (Bell, 1996). Coordinate System: NAD 1983 UTM 16N. Projection: Transverse Mercator."*

Figure 4: Image compression artifacts are visible in these vector diagrams. Please use little to no compression or vector format.

The image was changed for a higher resolution version.

Figure 7: Y-axis of (a): change to "Difference from the manual method". Properly reference the tables (Table S1 and S2 are gone)

Changes made as suggested. Reference to Table S1 had been changed to Table 2. Reference to Table S2 was eliminated.

---

## Author Response (AR3)

**Reply to Editor comments on**
*"An Estimate of Ice Wedge Volume for a High Arctic Polar Desert Environment, Fosheim Peninsula, Ellesmere Island"*
**by Claire Bernard-Grand'Maison and Wayne Pollard**

**General comments:**

Dear Ms. Bernard-Grand'Maison,

Thank you for your revisions. The reviewer's comments have been well incorporated into the revised version and its contributions and innovations are clear. I have a few minor corrections for you to consider before publication. Please see the detailed comments in the attached Comments to the Author.

Best regards,

Peter

Dear M. Morse,

Thank you for accepting our manuscript for publication in The Cryosphere. We have applied most of your suggested corrections. See detailed responses below.

Regards,

Claire Bernard-Grand'Maison and Wayne Pollard

**Specific comments:**

P1/L25: thermal protection from vegetation, a substantial surface organic soil layer, or thick snow cover. (Not just vegetation).

Changes made as suggested.

P2/L15: During the freezing season, rapid... / formation of cracks in frozen ground.

Changes made as suggested.

P5/L30: An Environment…

Changes made as suggested.

P5/L31-32 – P6/L1-2: The caption in Figure 3 refers to four surficial geology classes, but only one pooled class is shown to highlight where IW might be. Can you indicate in the Fig. caption that the surficial geology data are simplified, and here you can perhaps indicate the % area covered by the marine sediments that dominate. I think this way you can get away with not showing all of the surficial geology classes on the map. Perhaps say here: The area is mostly ... bedrock occur (Bell, 1996). Surficial geology of the area is dominated (???%) by unconsolidated ice-rich silty-clay marine sediments below ~150 m a.s.l., but local fluvial, glacial, and glaciofluvial deposits are present.

The caption of Figure 3 has been modified to indicate that the surficial geology data is simplified: *"Surficial geology data has been simplified and is from a map produced by Bell (1992)."*

The sentences in Section 2 (Study Area) have been changed following the suggestion. The dominance of the marine sediments is estimated to be ~ 60%. The text has been changed to: *"The surficial geology of the area is dominated (~60%)*

*by unconsolidated ice-rich silty-clay marine sediments below ~150 m a.s.l, but local fluvial, glacial, and glaciofluvial deposits are present."*

P6/L5: Change coldest to lowest and warmest to highest

Changes made as suggested.

P6/L8: Sparse vegetation (patchy low shrubs)

Changes made as suggested.

P14/L10: As with this earlier study…

Changes made as suggested.

P15/L18: our local observations relate directly to widespread permafrost thaw and development of thermokarst terrain… observed

Changes made as suggested.

P16/L9-13: Firstly, compared to manual delineation, two GIS-based semi-automated techniques – the Thiessen polygons methodology presented in Ulrich et al. (2014) and the Watershed Segmentation methodology, newly developed in this study – permit an acceptable approximation of IW volume in remote Arctic locations.

Changes made as suggested.

Figure 2 caption: Perhaps delete this text ("in a high-centered polygon environment"). Keeps things simple, and not tied to high or low centered environment, focuses on epigenetic IW, rather than state of polygons.

The authors agree to delete this text to keep it simple so the representation in a) can be interpreted by the reader.

Figure 3: Compression artifacts are evident on this Figure, especially when printed. Perhaps this is related to the PDF copy that I receive, but please ensure that the final figures are nice and crisp.

Compression artifacts are visible from the conversion of the document to PDF but the original figure is crisp.

Figure 3 caption: Perhaps delete this text as these classes are not shown on the map.

This text was deleted and the mention to the simplified geology classes was added. The caption is now changed to: "Surficial geology data has been simplified and is from a map produced by Bell (1992)."

Figure 4: Compression artifacts are evident on this Figure, especially when printed. Perhaps this is related to the PDF copy that I receive, but please ensure that the final figures are nice and crisp.

Compression artifacts are visible from the conversion of the document to PDF but the original figure is crisp.

**Technical comments (minor wording changes, typos, etc.):**

All the changes have been applied to the manuscript and improve its clarity.